# Eya3 partners with PP2A to induce c-Myc stabilization and tumor progression

Lingdi Zhang[1], Hengbo Zhou[2,3], Xueni Li[1], Rebecca L Vartuli[2,4], Michael Rowse[5], Yongna Xing[5], Pratyaydipta Rudra[6], Debashis Ghosh[6], Rui Zhao[1,4] & Heide L Ford[1,2,3,4]

Eya genes encode a unique family of multifunctional proteins that serve as transcriptional co-activators and as haloacid dehalogenase-family Tyr phosphatases. Intriguingly, the N-terminal domain of Eyas, which does not share sequence similarity to any known phosphatases, contains a separable Ser/Thr phosphatase activity. Here, we demonstrate that the Ser/Thr phosphatase activity of Eya is not intrinsic, but arises from its direct interaction with the protein phosphatase 2A (PP2A)-B55α holoenzyme. Importantly, Eya3 alters the regulation of c-Myc by PP2A, increasing c-Myc stability by enabling PP2A-B55α to dephosphorylate pT58, in direct contrast to the previously described PP2A-B56α-mediated dephosphorylation of pS62 and c-Myc destabilization. Furthermore, Eya3 and PP2A-B55α promote metastasis in a xenograft model of breast cancer, opposing the canonical tumor suppressive function of PP2A-B56α. Our study identifies Eya3 as a regulator of PP2A, a major cellular Ser/Thr phosphatase, and uncovers a mechanism of controlling the stability of a critical oncogene, c-Myc.

[1] Department of Biochemistry and Molecular Genetics, University of Colorado Denver Anschutz Medical Campus, Aurora 80045 CO, USA. [2] Department of Pharmacology, University of Colorado Anschutz Medical Campus, Aurora 80045 CO, USA. [3] Cancer Biology Program, University of Colorado Anschutz Medical Campus, Aurora 80045 CO, USA. [4] Molecular Biology Program, University of Colorado Anschutz Medical Campus, Aurora 80045 CO, USA. [5] McArdle Laboratory for Cancer Research, School of Medicine and Public Health, University of Wisconsin-Madison, Madison 53705 WI, USA. [6] Department of Biostatistics and Informatics, University of Colorado Anschutz Medical Campus, Aurora 80045 CO, USA. These authors contributed equally: Lingdi Zhang, Hengbo Zhou. Correspondence and requests for materials should be addressed to R.Z. (email: rui.zhao@ucdenver.edu) or to H..F. (email: heide.ford@ucdenver.edu)

The Eya proteins were initially discovered as factors required for normal eye development in *Drosophila*[1] and as essential co-activators of the Six family of homeoproteins[2–4]. Six and Eya proteins are critical regulators of embryonic development that control proliferation, survival, epithelial versus mesenchymal fates, and overall cell fate specification in numerous tissues[2,3,5]. Six1 and Eya are typically downregulated after organ development is complete, but are abnormally over expressed in multiple cancer types, significantly contributing to tumorigenesis and metastasis[6]. There are four Eya family members (Eya1–4) in mammals, each containing a highly conserved C-terminal Eya domain (ED) and a less conserved N-terminal transactivation domain (referred to as the N-terminal domain (NTD) or ED2). The ED mediates the Six1/Eya interaction, which is necessary for Six1 to promote breast cancer metastasis in mouse models[7,8]. The ED also contains protein Tyr phosphatase activity[2,3,9–12] that has been implicated in transcriptional activation[10], motility[13], DNA damage response[14,15], angiogenesis[16], and inhibition of the antitumor activity of estrogen receptor-β[17]. While the NTD is less conserved than the ED between different Eya family members (29–37% identity), conservation is high for a particular Eya family member between species (e.g., identity in the NTD of Eya3 amongst mammals is 97–100%)[5]. The NTD of all Eya proteins contains a P/S/T rich-transactivation domain[2]. Intriguingly, all Eya NTDs contain Ser/Thr phosphatase activity; yet this domain shows no homology to any known phosphatase motifs[18,19]. Because the Thr phosphatase activity of this domain is higher than the Ser phosphatase activity[18], we will refer to the activity as "Thr phosphatase" for the sake of simplicity. The Thr phosphatase activity of Eya has been implicated in innate immunity[18,20], and more recently in the dephosphorylation and stabilization of c-Myc during development[21,22], but its role in tumor initiation and/or progression has not been explored.

Eya proteins have an unexpected connection with protein phosphatase 2A (PP2A), the major Ser/Thr phosphatase in the cell[23,24], that will be addressed below. PP2A is a trimeric holoenzyme made up of a 65 kD structural subunit (A), a 36 kD catalytic subunit (C), and a regulatory subunit (B) of varying sizes. Mammals have two highly conserved A isoforms, Aα and Aβ (87% sequence identity). About 90% of PP2A holoenzymes contain the Aα isoform instead of the Aβ, which is predominantly expressed during embryogenesis. The catalytic subunit also has two highly conserved isoforms, Cα and Cβ (97% sequence identity), with Cα being the predominant isoform in most cells. Subunits A and C can interact and form the dimeric core enzyme, which only gains its full activity, subcellular localization, and substrate specificity after interacting with the regulatory B subunit to form the trimeric holoenzyme. B subunits are extraordinarily complex. At least 26 B subunits have been identified in humans, which can be grouped into four subfamilies: B (B55/PR55), B′ (B56/PR61), B″ (B72/PR72), and B‴ (PR93/PR110). Each subfamily contains several isoforms (e.g., B has α, β, γ, and δ isoforms), some of which have several splice variants. Unlike A and C subunits, the sequences and structures of the B subunits are highly varied, contributing to the diverse localization, substrate specificity, and function of PP2A holoenzymes[23,24].

PP2A regulates many essential cellular processes, including migration, transformation, DNA replication, translation, apoptosis, and stress response regulation, as well as cell cycle progression and metabolism[23,25]. To modulate these processes, PP2A acts on numerous signaling pathways by dephosphorylating key components of the pathways. The particular function of PP2A is often controlled by the specific regulatory subunit it contains. For example, the B56α subunit directs PP2A to c-Myc, whose dephosphorylation by PP2A at residue S62 leads to c-Myc degradation by the ubiquitin pathway[26]. This function of PP2A, along with others, is generally thought to make PP2A a tumor suppressor[23]. Indeed, mutations that inactivate PP2A, loss of PP2A activators, and overexpression of PP2A inhibitors all contribute to tumorigenesis[27]. However, PP2A can be tumor promotional, depending on the context. For example, PP2A can inhibit apoptosis by dephosphorylating Bcl-2 in tumor cell lines[28], and it can also destabilize p53 by dephosphorylation[29]. Furthermore, in colorectal tumor cells, pT239 of c-Jun can be dephosphorylated by the B55α-associated PP2A complex, leading to increased target gene occupancy by c-Jun, and increased migration and invasion[30]. Finally, a recent study has shown that PP2A-B55α is essential for pancreatic cancer progression via maintaining heightened AKT, ERK, and Wnt oncogenic signaling[31].

In this paper, we demonstrate that Eya's Thr phosphatase activity is not intrinsic, as has been previously reported[19–21], but comes from its association with the PP2A. We show that the NTD of Eya proteins directly interacts with the B55α subunit of PP2A. We further demonstrate that Eya3 regulates the stability of c-Myc not via an intrinsic Thr phosphatase activity, but rather by controlling the dephosphorylation of c-Myc at pT58 through PP2A, suggesting that Eya3 may change the function of PP2A from tumor suppressive to tumor promotional. In line with a tumor promotional role of the Eya3–PP2A complex, we demonstrate in mouse models of breast cancer metastasis that Eya3 that is unable to interact with B55α has reduced metastasis, and that knockdown (KD) of the B55α subunit inhibits metastasis. Our results reveal a previously unrecognized function of Eya in regulating PP2A activity, which in turn influences c-Myc stability, as well as tumor progression.

## Results

**Eya3 associates with the Ser/Thr phosphatase PP2A.** Despite the absence of any recognizable phosphatase motif in their N-terminal domains, all Eya proteins are reported to contain an N-terminal Thr phosphatase activity, and mouse Eya3 (mEya3) has been shown to have the highest activity among Eya family members[18]. Thus, we focused on Eya3 to determine the underlying mechanism for its Thr phosphatase activity. To this end, we expressed and purified mouse Eya3 (87% sequence identity to the full-length human Eya3 and 97% identity to the human Eya3 NTD) which contains the Thr phosphatase activity[19], from *Escherichia coli* (E-Eya3) and mammalian HEK293FT cells (M-Eya3) (Fig. 1a). We found that Eya3 purified from mammalian cells has robust Thr and Tyr phosphatase activities, and very weak Ser phosphatase activity, as measured using a malachite-green-based phosphatase assay with a phospho-Thr (KRpTIRR), phospho-Tyr (pH2AX, KATQASQEpY), or phospho-Ser (RRApSVA)-containing peptides as substrates (Fig. 1b and c, Supplementary Fig. 1a), consistent with previous reports[18,21]. Since the Thr phosphatase activity of Eya3 is much higher than its Ser phosphatase activity (Supplementary Fig. 1a), we focused on the Eya3 Thr phosphatase activity in all subsequent experiments. In contrast, Eya3 expressed and purified from *E. coli* has robust Tyr phosphatase activity (indicating that the protein is well-folded), but has no detectable Thr phosphatase activity (Fig. 1b and c). This observation, coupled with the absence of any recognizable phosphatase motifs, raised the possibility that the observed Ser/Thr phosphatase activity of Eya3 is not intrinsic, but arises from proteins associated with Eya3 purified from the mammalian system.

To identify potential phosphatases associated with Eya3, we analyzed Eya3 purified from HEK293FT cells using mass spectrometry, which revealed the presence of Aα, Aβ, B55α, Cα, and Cβ subunits of PP2A, in addition to Eya3 (Table 1). We

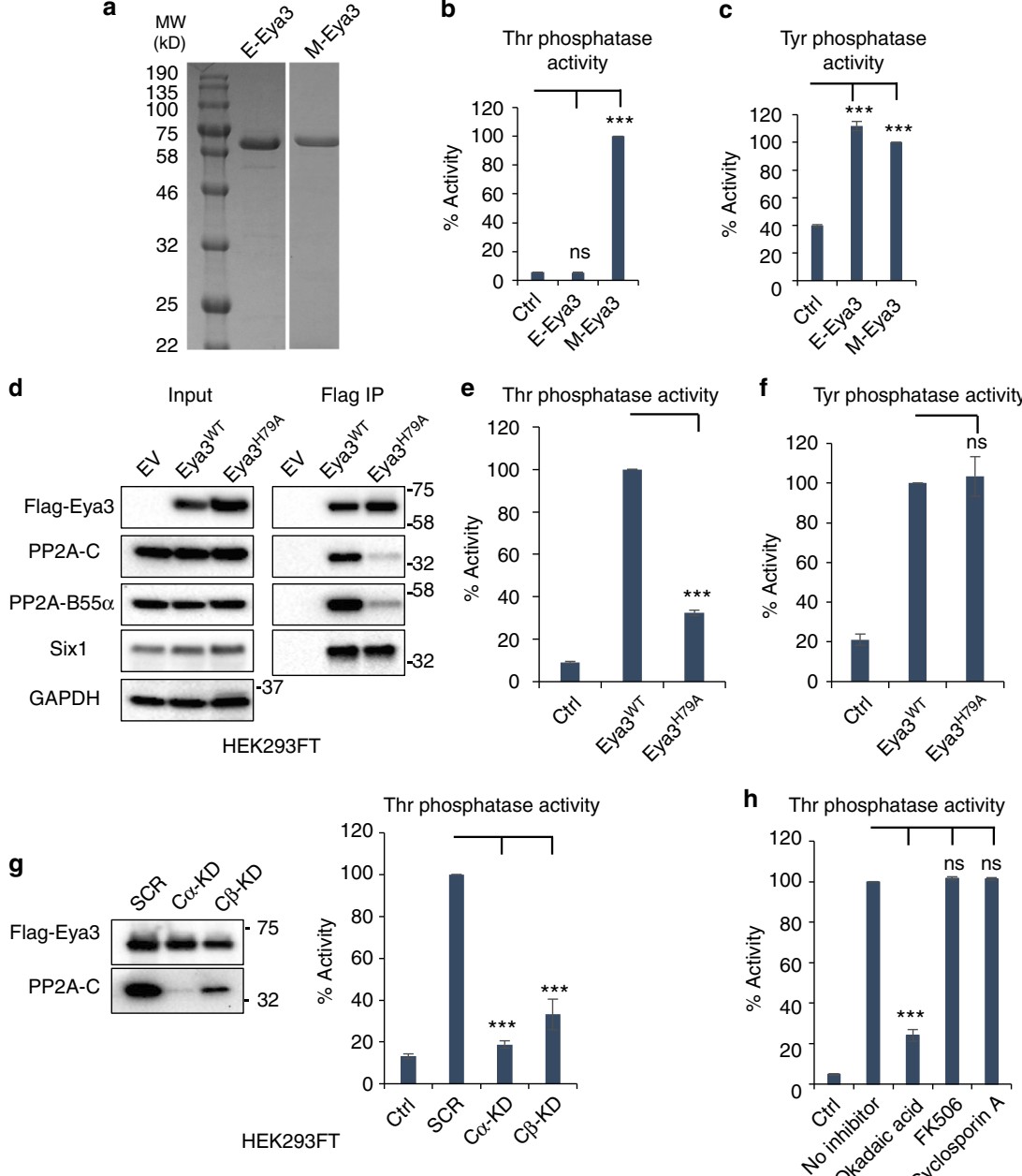

**Fig. 1** The Thr phosphatase activity of Eya3 arises from its association with PP2A. **a** SDS PAGE gel shows mEya3 purified from *E. coli* (E-Eya3) and mammalian HEK293FT cells (M-Eya3). **b** Thr and **c** Tyr phosphatase activities of E-Eya3 and M-Eya3 were analyzed in malachite-green-based phosphatase assays using phosphor-Thr (KRpTIRR) or phosphor-Tyr (pH2AX, KATQASQEpY) peptide as substrates. All phosphatase activities were normalized to the amount of protein used in the assay. The phosphatase activity of M-Eya3 was set to 100%. All control reactions (Ctrl) in Fig. 1 refer to samples in which no protein was added. **d** Immunoprecipitation (IP) using a Flag antibody was conducted on lysates of HEK293FT cells transfected with empty vector (EV), Flag-Eya3 (Eya3[WT]) or Flag-H79A mutant Eya3 (Eya3[H79A]). Western blot analysis of IPs shows that the H79A mutation diminishes the interaction of Eya3 with PP2A, but does not affect the interaction of Eya3 with Six1. Molecular weight markers in kilodalton are labeled on the side of western blots. **e** Thr and **f** Tyr phosphatase activities of Eya3[WT] and Eya3[H79A] were evaluated using Eya3 immunoprecipitated from HEK293FT cells. Eya3[WT] phosphatase activity was set to 100%. **g** Thr phosphatase activity assay was conducted using Eya3 immunoprecipitated from scramble control (SCR), PP2A-Cα, or PP2A-Cβ KD HEK293FT cells. Left: IP using a Flag antibody followed by western blot analysis shows that similar levels of Eya3 protein were obtained from HEK293FT cells transduced with scramble shRNA, PP2A-Cα shRNA, or PP2A-Cβ shRNA, whereas Eya3 associates with less PP2A when PP2A-Cα or PP2A-Cβ is KD. Right: Thr phosphatase assay using Eya3 proteins shown in left panel. Phosphatase activity for SCR group was set to 100%. **h** Thr phosphatase activity assay was performed using M-Eya3 in the presence of DMSO (no inhibitor), PP2A inhibitor okadaic acid (2.5 nM), PP2B inhibitor FK506 (5 μM), or calcineurin inhibitor cyclsoporin A (5 μM). The no inhibitor group was set to 100% phosphatase activity. Data shown are mean ± SD from three independent experiments. In the phosphatase activity assay, the *p*-value is calculated using one-way ANOVA followed by Tukey test, \*\**p* < 0.01, and \*\*\**p* < 0.001

**Table 1 Mass spectrometry analysis identified multiple PP2A subunits associated with mEya3 purified from HEK293FT cells**

| Protein | Total spectrum counts |
| --- | --- |
| PP2A-Aα | 67 |
| PP2A-Aβ | 33 |
| PP2A-B55α | 44 |
| PP2A-Cα | 37 |
| PP2A-Cβ | 37 |

confirmed the association of PP2A with Eya3 using immunoprecipitation (IP) of Flag-tagged Eya3 from HEK293FT cells followed by western blot analysis with anti-PP2A-C and anti-PP2A-B55α antibodies (Fig. 1d). We further demonstrated that the previously described Thr phosphatase-dead Eya3[H79A], Eya3[Y77A], Eya3[Y90A], and Eya3[Y77AH79A] mutants[19] all have significantly reduced "Thr phosphatase" activity (Fig. 1e and Supplementary Fig. 2a) and reduced association with PP2A (Fig. 1d and Supplementary Fig. 2b). Importantly, the Eya3[H79A] mutant that we used throughout our study does not have a reduced association with Six1 (Fig. 1d), nor does the mutation inhibit transcriptional activity of the Six1/Eya complex using a MEF3 promoter-luciferase reporter[32] (Supplementary Fig. 2c). Further, as anticipated, the H79A mutation in Eya3 does not interfere with its Tyr phosphatase activity (Fig. 1f), which is localized to the C-terminal domain (CTD) of the protein[9,10,14]. To confirm that the Thr phosphatase activity of Eya3 is dependent on its association with PP2A, we performed KD of PP2A-Cα or Cβ (KD levels shown in Supplementary Fig. 3a), and found that KD of either catalytic subunit of PP2A significantly reduced the Thr phosphatase activity of Eya3 (Fig. 1g). In addition, okadaic acid, a PP2A inhibitor, inhibits the majority of the Thr phosphatase activity of Eya3 when used at 2.5 nM, a concentration which specifically inhibits PP2A, but not other Ser/Thr phosphatases, such as PP1[33]. FK506 and cyclosporin A, which inhibit calcineurin with IC50s of 0.5 and 5 nM, respectively[34], and both of which inhibit PP2B with an IC50 of ~30 nM[35], do not have any effect on the Thr phosphatase activity of Eya3 at 5 μM (Fig. 1h).

We next evaluated whether Eya3 and PP2A associate in the cell. Using the proximity ligation assay (PLA) in HEK293FT cells transfected with either wildtype (WT) or H79A mutant Eya3 (Fig. 2a), we demonstrate that Eya3[WT], but not Eya3[H79A], associates with PP2A (Fig. 2b and c). Furthermore, using the same assay, we show that endogenous Eya3 and PP2A associate in multiple human and mouse cell lines, including both embryonic and cancerous cell lines (Supplementary Fig. 4a and Fig. 2d). Taken together, these data strongly suggest that the "Thr phosphatase activity" of Eya3 arises from its association with the prototypic cellular Thr phosphatase PP2A.

**Eya3 associates with PP2A through its NTD.** We demonstrate that human Eya1–4 purified from HEK293FT cells associate with PP2A and have Thr phosphatase activity (Fig. 3a and b). Although a previous report shows that mEya3 has the highest Thr phosphatase activity among mouse Eya family members[19], our data demonstrate that human Eya2, while expressed less efficiently than the other human Eyas, has greater binding affinity to PP2A and thus has the highest Thr phosphatase activity of human Eyas when the activity is normalized to protein levels used in the assay (Fig. 3b).

Previous studies demonstrated that the NTD of Eya family members contains the Thr phosphatase activity[19]. Thus, to

decipher which domain of mEya3 interacts with PP2A, we expressed and purified the NTD (residues 1–240) and the CTD (residues 241–526) of Eya3 from HEK293FT cells. IP of full-length Eya3 and the various Eya3 domains, followed by measurement of Thr phosphatase activity, demonstrates that the NTD of Eya3 predominantly interacts with PP2A and has significant Thr phosphatase activity (Fig. 3c and d), while the CTD has little association with PP2A (Fig. 3c), and dramatically reduced phosphatase activity compared to the WT (to a level similar to the control) (Fig. 3d). These data again support the finding that Eya3 acts as a Thr phosphatase via its association with PP2A, and further localize this interaction and activity to its NTD.

**Eya3 directly interacts with the B55α subunit of PP2A.** Examination of our mass spectrometry analyses of Eya3 purified from HEK293FT cells suggested that Eya3 interacts directly with a B subunit of PP2A. Although mammals have a large number of different regulatory B subunits, B55α was the only B subunit identified in our mass spectrometry analysis of proteins associated with purified Eya3, along with multiple A (Aα and Aβ) and C (Cα and Cβ) subunits (Table 1). These data suggest that Eya3 directly interacts with B55α, rather than the A and C subunits that form the common core associated with all PP2A regulatory B subunits. To further test this hypothesis, we expressed and purified GST-Aα from E. coli, as well as His-B55α and His-Cα from insect cells (B55α and Cα are either insoluble or not functional when expressed in E. coli)[36–38]. The PP2A holoenzyme assembled using these purified proteins has both Ser/Thr phosphatase activities (Supplementary Fig. 1b), suggesting that these purified subunits are well-folded and fully functional. Using purified proteins, we demonstrated that His-B55α alone or GST-Aα + His-B55α + His-Cα together can pull down purified Flag-Eya3, while GST-Aα alone or GST-Aα + His-Cα cannot; demonstrating that B55α is necessary and sufficient for interacting with Eya3 (Fig. 4a). GST-B56γ1, on the other hand, does not demonstrate any interaction with Eya3 in a pull-down experiment (Fig. 4b).

We next evaluated the association between Eya3 and B55α in cells using PLA experiments. We demonstrate that KD of B55α, as expected, diminishes the interaction of the B55α regulatory subunit of PP2A with Eya3 in 66cl4 mammary carcinoma cells, demonstrating the specificity of our PLA experiments (Supplementary Figs 3b and 4b,c). We further demonstrate that KD of B55α abolishes the association of the catalytic subunit of PP2A with Eya3 overexpressed in HEK293FT cells (Fig. 4c–e), as well as with endogenous Eya3 in 66cl4 mammary carcinoma cells[39] (Fig. 4f–h; Supplementary Fig. 4d,e shows an additional B55α KD) using anti-PP2A-C and Eya3 antibodies. Finally, Eya3 appears to specifically interact with the B55α subunit of PP2A, as we could not detect an association between B56α and Eya3 in 66cl4 cells (scramble control (SCR) or B55α KD, Supplementary Fig. 5b), in line with our in vitro observation that Eya3 does not interact with the PP2A-B56γ1 subunit either in pull-down experiments (Fig. 4b). Taken together, our data demonstrate that Eya3 physically interacts with the PP2A holoenzyme through the B55α subunit, mediating the Thr phosphatase activity of Eya3.

**Eya3–PP2A regulates c-Myc stability.** We next set out to evaluate whether the Eya3-associated Thr phosphatase activity, via PP2A, regulates c-Myc stability. While it has not been examined before for Eya3, it was previously shown that the "intrinsic" Thr phosphatase activity of Eya1 is critical for dephosphorylating pT58 of c-Myc, thereby preventing the dephosphorylation of pS62 by PP2A-B56α and leading to increased c-Myc stability[21,22]. Consistent with the possibility that Eya3-associated PP2A

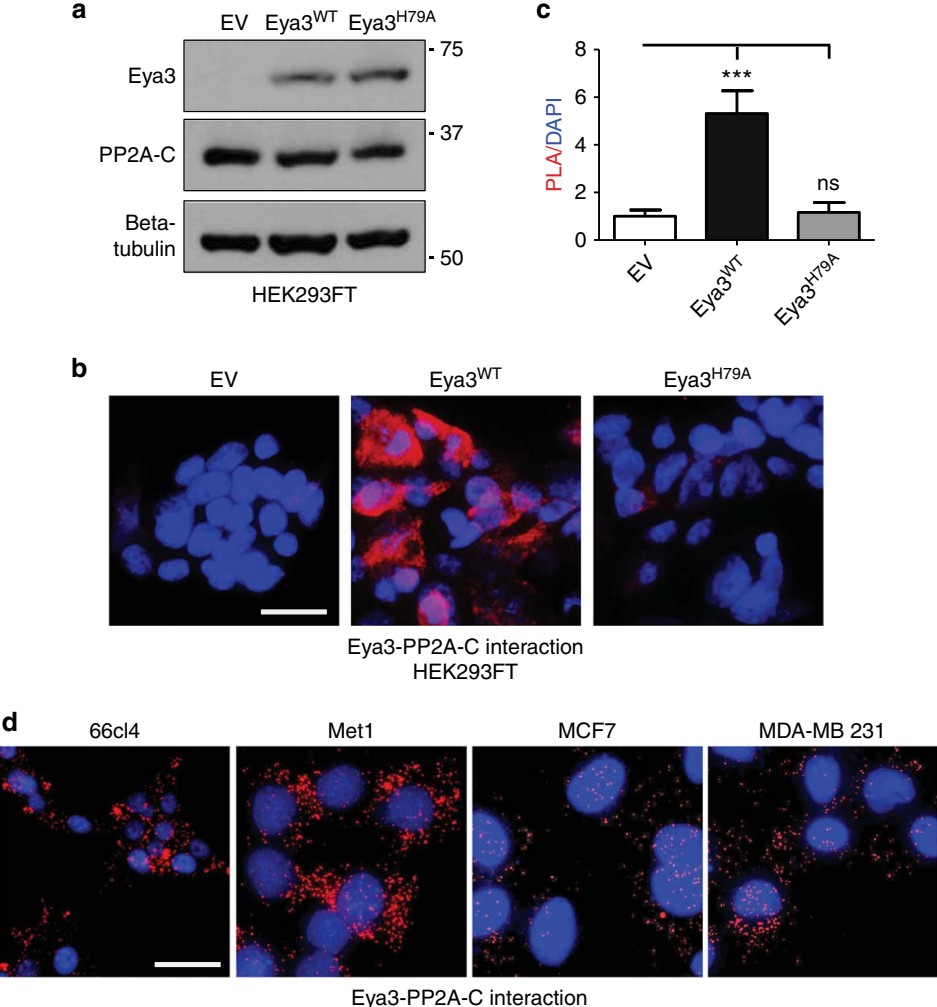

**Fig. 2** The Eya3–PP2A association is disrupted by the Eya3 H79A mutation in cells, and can be observed in the endogenous context. **a** Western blot analysis on whole cell lysates demonstrates levels of exogenous Eya3, as detected with an anti-Eya3 antibody, expressed in HEK293FT cells transfected with empty vector (EV), Flag-Eya3$^{WT}$ or Flag-Eya3$^{H79A}$. Molecular weight markers in kilodalton are labeled on the side of western blots. **b** Representative images of proximity ligation assay (PLA) demonstrating the association between Eya3 and PP2A-C in HEK293FT cells transfected with EV, Eya3$^{WT}$, or Eya3$^{H79A}$ are shown, scale bar: 20 μm. **c** Quantification of relative PLA/DAPI signal ratio in **b**, EV group is set to 1 and $p$-value is calculated using one-way ANOVA followed by Tukey test, ***$p < 0.001$, error bars: standard deviations (SD), $n = 5$. **d** Representative PLA images examining the endogenous Eya3–PP2A-C association in multiple mouse and human breast cancer lines: 66cl4, Met1, MCF7, and MDA-MB-231 are shown, scale bar: 20 μm

phosphatase also dephosphorylates c-Myc, we demonstrated that Eya3 expressed in HEK293FT cells is associated with c-Myc in an IP experiment and that the H79A mutation did not affect the association between Eya3 and c-Myc (Fig. 5a).

To determine whether Eya3 can indeed dephosphorylate c-Myc at pT58, and is thereby stabilizing c-Myc via Eya3′s association with PP2A, we first generated Eya3 addback lines using the triple negative mouse mammary carcinoma cell line, 66cl4[39]. We chose the 66cl4 triple negative breast cancer (TNBC) system, due to previous evidence that Eya3 plays a role in the metastasis of TNBC[13]. We performed Eya3 KD in 66cl4 cells, then reintroduced into the Eya3 KD cells empty vector (EV), Flag-Eya3$^{WT}$, or Flag-Eya3$^{H79A}$, the latter two carrying wobble mutations to avoid KD by the shRNA construct targeting endogenous Eya3 within the cells. In total, four cell lines were generated, which will be referred to as SCR + EV (control line), KD + EV, KD + Flag-Eya3$^{WT}$, and KD + Flag-Eya3$^{H79A}$. We performed western blot analyses of these cell lines, confirming that addback of Flag-tagged Eya3$^{WT}$ or Eya3$^{H79A}$ restored Eya3 to levels comparable to the endogenous level of Eya3 (Supplementary Fig. 6a, input

panel). As observed using IP (Supplementary Fig. 6a, IP panel) and PLA (Supplementary Fig. 6b–e), introduction of Flag-Eya3$^{WT}$ in Eya3 KD cells restored the Eya3–PP2A interaction (both PP2A-C and B55α) in cells, whereas introduction of Flag-Eya3$^{H79A}$ did not significantly restore this interaction, consistent with our earlier observation that the H79A mutation abolishes the B55α–Eya3 interaction (Fig. 1d).

To determine whether the Eya3-associated Thr phosphatase controls the stability of c-Myc, we examined the levels of pT58 c-Myc (a form subject to degradation)[40] and pS62 c-Myc in the 66cl4-Eya3 KD and addback lines. Similar to Eya1[21,22], loss of Eya3 increased the level of the pT58 c-Myc (Fig. 5b). Addback of Flag-Eya3$^{WT}$, but not Flag-Eya3$^{H79A}$, led to dephosphorylation of pT58 on c-Myc in Eya3 KD cells (Fig. 5b). In contrast, pS62 c-Myc levels were increased in the presence of endogenous Eya3 or Flag-Eya3$^{WT}$. Similarly, total c-Myc levels were increased only in the presence of endogenous Eya3 or Flag-Eya3$^{WT}$ (Fig. 5b). To further determine whether the association of Eya3 with PP2A-B55α is important for stabilization of c-Myc through dephosphorylation of pT58 specifically, we knocked down the B55α

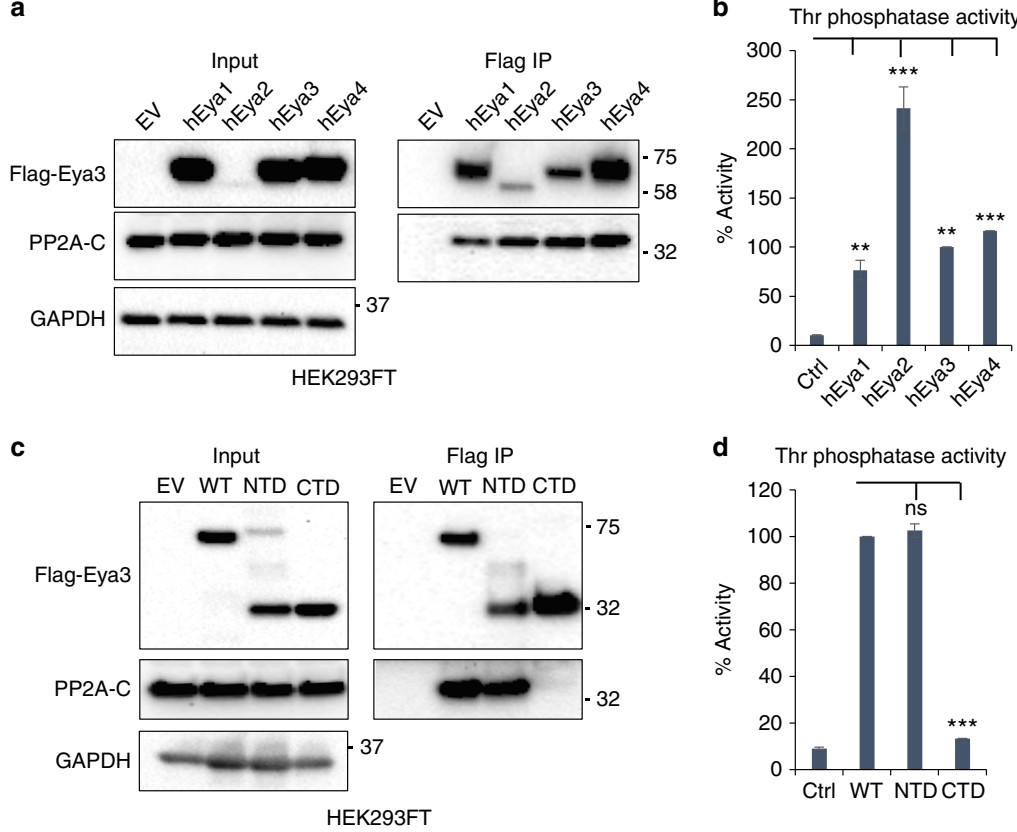

**Fig. 3** All Eya family members interact with PP2A, and the N-terminal domain (NTD) of Eya3 is the main contributor to the PP2A interaction. **a** HEK293FT cells were transfected with EV, Flag-fused human Eya1 (hEya1), hEya2, hEya3, or hEya4. IP against Flag demonstrates the levels of PP2A-C subunit associated with Flag-hEyas. Molecular weight markers in kilodalton are labeled on the side of western blots. **b** Thr phosphatase activity assay was conducted using hEyas immunoprecipitated from transfected HEK293FT cells. All phosphatase activity was normalized to the amount of protein used in the assay. hEya3 Thr phosphatase activity was set to 100%. **c** HEK293FT cells were transfected with EV, Flag-fused wildtype (WT), NTD, or C-terminal domain (CTD) Eya3. IP against Flag demonstrates the levels of PP2A-C subunit associated with different domains of Flag-Eya3. **d** Thr phosphatase activity assay was conducted using different domains of Eya3 immunoprecipitated with a Flag antibody from transfected HEK293FT cells. Eya3 WT Thr phosphatase activity was set to 100%. Data shown are mean ± SD from three independent experiments. In the phosphatase activity assay, the $p$-value is calculated using one-way ANOVA followed by Tukey test, $**p < 0.01$, and $***p < 0.001$

subunit of PP2A in 66cl4 cells, which led to a marked increase in levels of pT58 on c-Myc (Supplementary Fig. 7a). However, we did not observe significant alterations in pS62 or total c-Myc at steady state with B55α KD (Supplementary Fig. 7a), suggesting that the loss of B55α only partially phenocopies the loss of Eya3.

To better examine differences in total c-Myc levels, we measured the half-life of c-Myc in the presence or absence of Eya3 or its associated Thr phosphatase activity, using a cycloheximide chase assay in HEK293FT cells (Eya3 expression levels in these cells are shown in Supplementary Fig. 7b). The presence of Flag-Eya3$^{WT}$ increased the half-life of c-Myc, as expected, when compared to either the absence of Eya3 (EV control) or addback of Flag-Eya3$^{H79A}$ (Fig. 5c, quantified in Fig. 5d). Importantly, expression of Eya3 in the presence of PP2A-B55α increased the c-Myc half-life in HEK293FT cells (Eya3 and PP2A-B55α expression levels shown in Supplementary Fig. 7c), whereas Eya3 could not stabilize c-Myc in the absence of B55α (Fig. 5e; quantified in 5f). Thus, our data suggest that Eya3 controls c-Myc levels through dephosphorylating T58 via a mechanism that requires PP2A-B55α.

**Eya3–PP2A Thr phosphatase regulates late-stage metastasis.**
We demonstrated that the Eya3-associated Ser/Thr phosphatase activity is mediated by PP2A, rather than being intrinsic, and is

important for c-Myc stability. While the role of c-Myc in metastasis is controversial[41], it is generally believed to be pro-metastatic[42–45]. Moreover, a previous study has implicated Eya3 in breast cancer metastasis, likely at least in part through additional means, such as regulation of the actin cytoskeleton[13]. Although only the Tyr phosphatase activity of Eya3 has previously been implicated in metastasis[13], our data suggest that the Thr phosphatase activity of Eya3 may also impinge on this process. To test this hypothesis, we performed experimental metastasis assays by injecting $2.5 \times 10^6$ of SCR + EV, KD + EV, KD + Flag-Eya3$^{WT}$, or KD + Flag-Eya3$^{H79A}$ 66cl4 cells into the tail vein of female BALB/c immune-competent mice. Consistent with previous work, Eya3 KD decreased late-stage metastatic burden. Strikingly, metastasis was rescued in Eya3 KD cells by adding back Flag-Eya3$^{WT}$, but not Thr phosphatase-dead Flag-Eya3 (Eya3$^{H79A}$) (Fig. 6a, b). As expected, a higher percentage of c-Myc positive regions were found in lung metastases from SCR + EV and KD + Flag-Eya3$^{WT}$, when compared to KD + EV and KD + Flag-Eya3$^{H79A}$, demonstrating that the Eya3-associated Thr phosphatase activity stabilized c-Myc throughout the metastatic process (Fig. 6c, d). Mice injected with KD + EV or KD + Flag-Eya3$^{H79A}$ cells also exhibited greater survival, compared to mice injected with SCR + EV or KD + Flag-Eya3$^{WT}$ cells (Fig. 6e). Because the H79A mutation disrupts the Eya3 interaction with the B55α subunit of PP2A (Fig. 1d, Supplementary

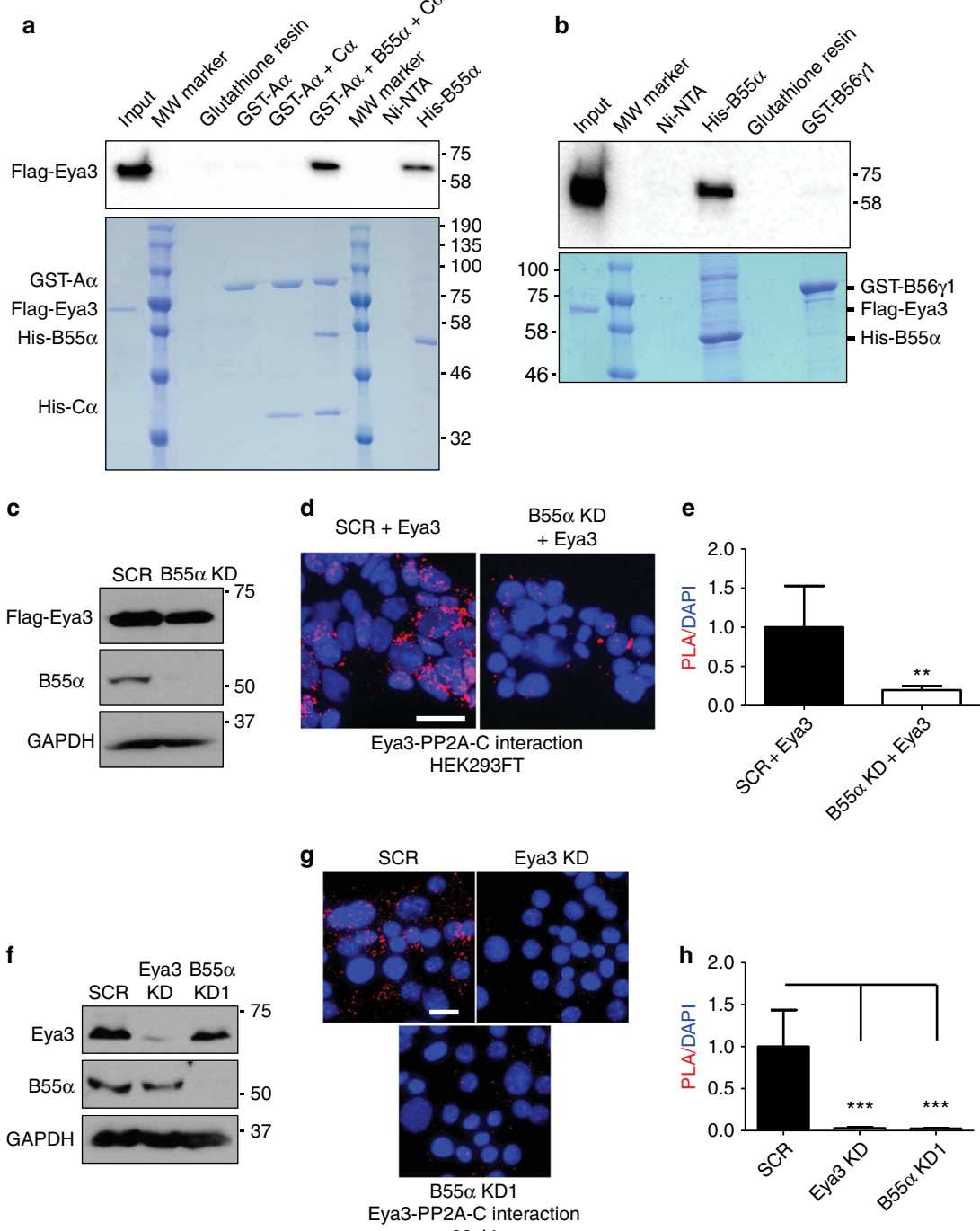

**Fig. 4** Eya3 interacts directly with the PP2A-B55α subunit. **a** Top: GST-Aα, GST-Aα + B55α, GST-Aα + B55α + Cα, or His-B55α were used to pull down purified Flag-Eya3, which was identified using an anti-Flag antibody and western blot analysis. Bottom: Coomassie blue stained SDS PAGE gel demonstrating the amount of PP2A subunits used for pull-down experiments. **b** His-B55α and GST-B56γ1 were used to pull down purified Flag-Eya3, which is identified using an anti-Flag antibody and western blot analysis. Bottom: Coomassie blue stained SDS-PAGE gel demonstrating the amount of PP2A subunits used for pull-down experiments. **c** Western blot analysis shows protein levels of Flag-Eya3 and PP2A-B55α in SCR or PP2A-B55α KD HEK293FT cells transfected with Flag-Eya3. Molecular weight markers in kilodalton are labeled on the side of western blots. **d** Representative images of PLA demonstrate the Eya3–PP2A-C interaction in SCR or PP2A-B55α KD HEK293FT cells transfected with Eya3, scale bar: 10 μm. **e** Quantification of PLA/DAPI signal ratio in **d**, SCR group is set to 1 and $p$-value is calculated using an unpaired Student's $t$-test, **$p < 0.01$, error bars: standard deviations (SD), $n = 5$. **f** Western blot analysis shows endogenous protein levels of Eya3 and PP2A-B55α in SCR, Eya3 KD, or PP2A-B55α KD 66cl4 cells. **g** Representative images of PLA demonstrate the Eya3–PP2A-C association in SCR, but not in Eya3 KD or PP2A-B55α KD 66cl4 cells, scale bar: 10 μm. **h** Quantification of PLA/DAPI signal ratio in **g**, SCR group is set to 1 and $p$-value is calculated using the one-way ANOVA followed by Tukey test, ***$p < 0.001$, error bars: SD, $n = 5$

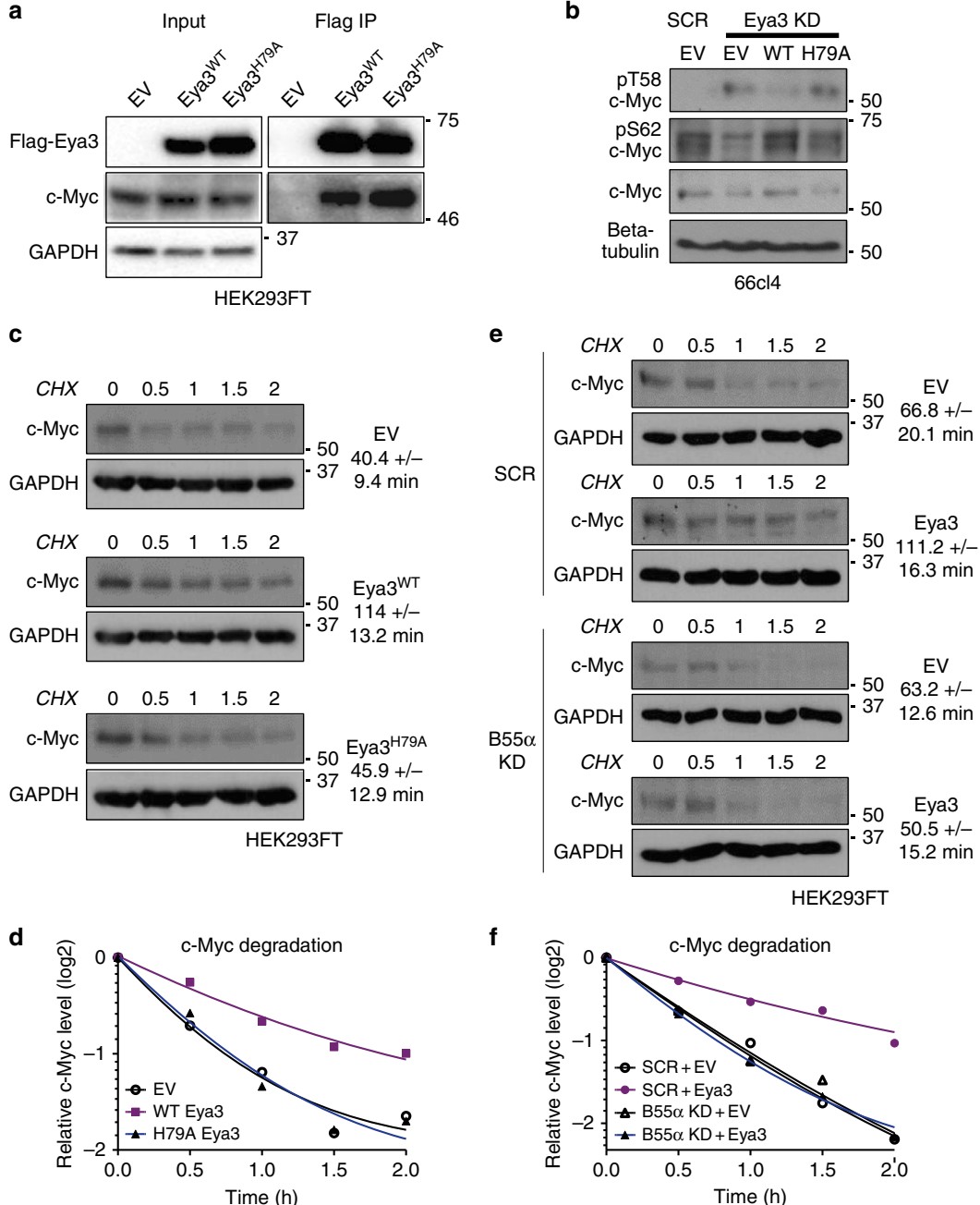

**Fig. 5** Eya3–PP2A-associated Thr phosphatase activity increases c-Myc stability. **a** IP from HEK293FT cells using a Flag antibody demonstrates Eya3[WT] and Eya3[H79A] both associated with c-Myc. Molecular weight markers in kilodalton are labeled on the side of western blots. **b** Western blot analysis shows pT58, pS62, and total c-Myc levels in SCR + EV, Eya3 KD + EV, Eya3 KD + WT, and Eya3 KD + H79A 66cl4 cells. **c** Cycloheximide chase assays were conducted using HEK293FT cells transfected with EV, Eya3[WT], or Eya3[H79A] to measure c-Myc stability. Representative images of western blot analyses examining c-Myc levels in HEK293FT cells at various treatment time points are shown. **d** Quantification of relative c-Myc levels over the treatment time course, with exponential nonlinear regression applied for modeling, $n \geq 4$. **e** Cycloheximide chase assays were conducted using SCR and B55α KD HEK293FT cells transfected with EV or Eya3[WT] to measure c-Myc stability. **f** Quantification of relative c-Myc levels during the treatment time course, $n = 5$

Fig. 6b,c), the difference between Eya3[H79A] and Eya3[WT] suggests that B55α could act to promote tumor progression, rather than suppress it, a property normally attributed to PP2A[24,46]. Indeed, when compared to the control group, KD of B55α in 66cl4 cells reduced the metastatic burden (Fig. 7a, b) and extended the survival of the mice (Fig. 7c). Collectively, these data indicate that the Eya3-associated Thr phosphatase activity is essential for its role in late-stage metastasis via interaction with PP2A.

## Discussion

It has long been thought that the Eya proteins possess intrinsic Tyr and Thr phosphatase activities through two different domains on the same protein[19]. Although the C-terminal ED, which carries the Tyr phosphatase activity, contains the signature motif for the haloacid dehalogenase (HAD) phosphatase superfamily, the NTD of Eya, which carries the Thr phosphatase activity, contains no identifiable phosphatase motifs[19]. We

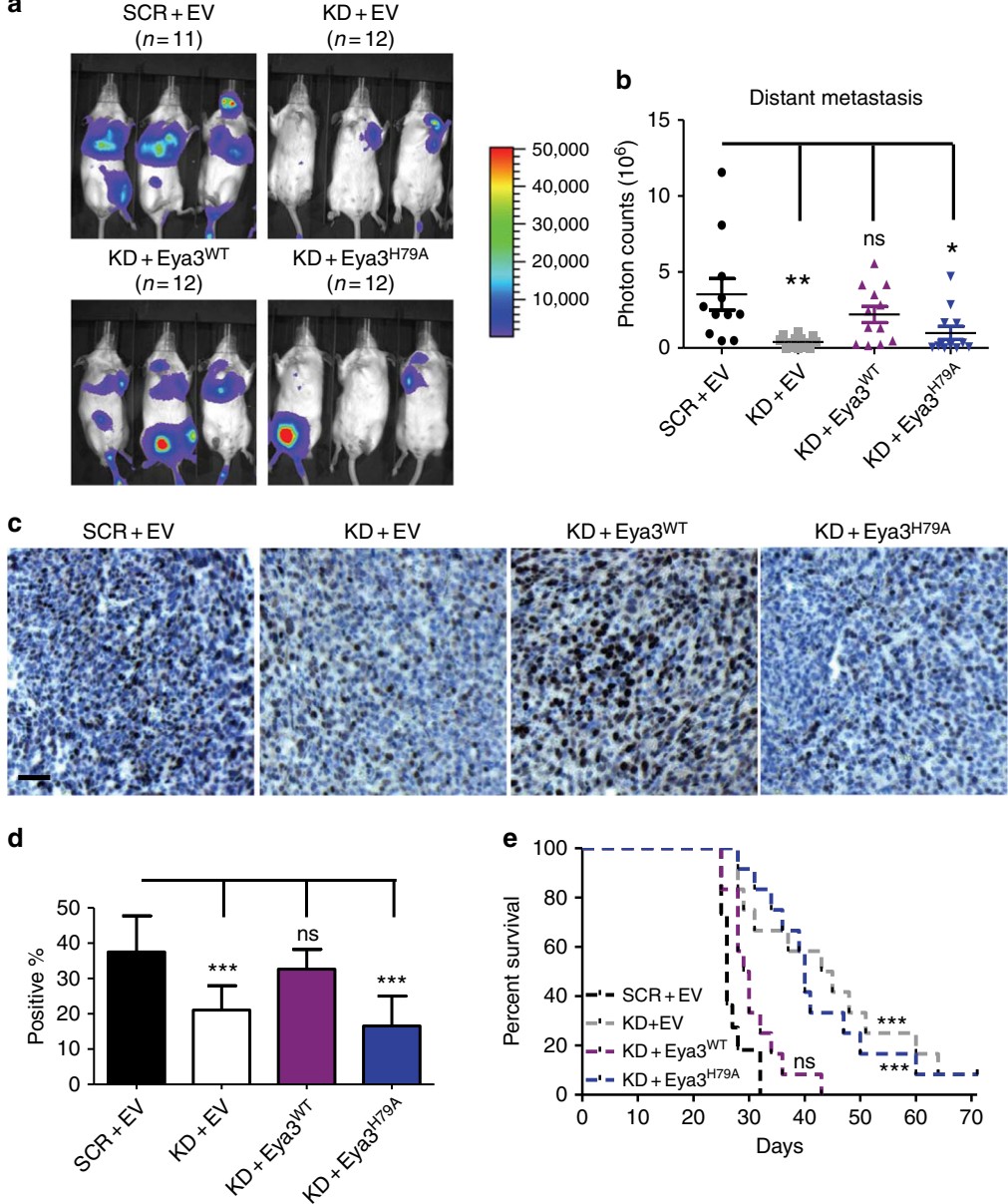

**Fig. 6** Eya3–PP2A-associated Thr phosphatase activity increases breast cancer metastasis and positively correlates with the percentage of c-Myc-positive regions in lung metastases. Experimental metastasis assay was performed by injecting BALB/c mice with firefly luciferase-tagged 66cl4 cells as follows: SCR + EV, KD + EV, KD + Eya3$^{WT}$, or KD + Eya3$^{H79A}$. **a** Representative bioluminescent images of metastatic burden in mice at week 3 after injection. **b** Quantification of photon flux in **a**, p-value is calculated using one-way ANOVA followed by Tukey test, *$p < 0.05$, **$p < 0.01$. **c** Representative immunohistochemical staining of lung metastases using a c-Myc antibody. Note that the percentage of c-Myc positive cells is highest in the presence of WT Eya3 (either endogenous or addback), and is diminished with Eya3 KD or addback of the Eya3$^{H79A}$ mutant. **d** Graph shows the quantification of c-Myc positive regions across different conditions. Percentage of positive cells was assessed using ImageJ software. p-value was calculated using one-way ANOVA followed by Tukey test, ***$p < 0.001$, error bars: standard deviation. 20 metastases from each condition were examined, scale bar: 20 μm. **e** Kaplan–Meier curves demonstrate the percentage of mice that survived until day 70. p-value is calculated using the log-rank test followed by Bonferroni correction, ***$p < 0.001$

demonstrate in this paper that, contrary to the previous paradigm, the Thr phosphatase activity of Eya3 (and other Eya family members) is not intrinsic, but is rather mediated by its interaction with PP2A, the prototypic Ser/Thr phosphatase in cells. Previous efforts made to investigate the mechanisms by which Eya proteins function as intrinsic Thr phosphatases identified a number of conserved residues (C56, Y77, H79, and Y90) in Eya3 that are important for its Thr phosphatase activity[19]. We demonstrate that mutating these residues (Y77A, H79A, or Y90A) disrupt the interaction between Eya3 and PP2A, resolving the conundrum of

the lack of any recognizable phosphatase motif in the NTD of Eya.

Our findings in this paper underscore the observation that Eya is a protein with diverse functions fulfilled by its different domains. The NTD of Eya carries out the transactivation function and interacts with PP2A. The CTD of Eya is responsible for interacting with Six1 and contains the HAD-family Tyr phosphatase activity. The transcriptional activation function of Eya is clearly important for many Six1-mediated tumorigenic and metastatic properties[47]. The Tyr phosphatase activity also plays a

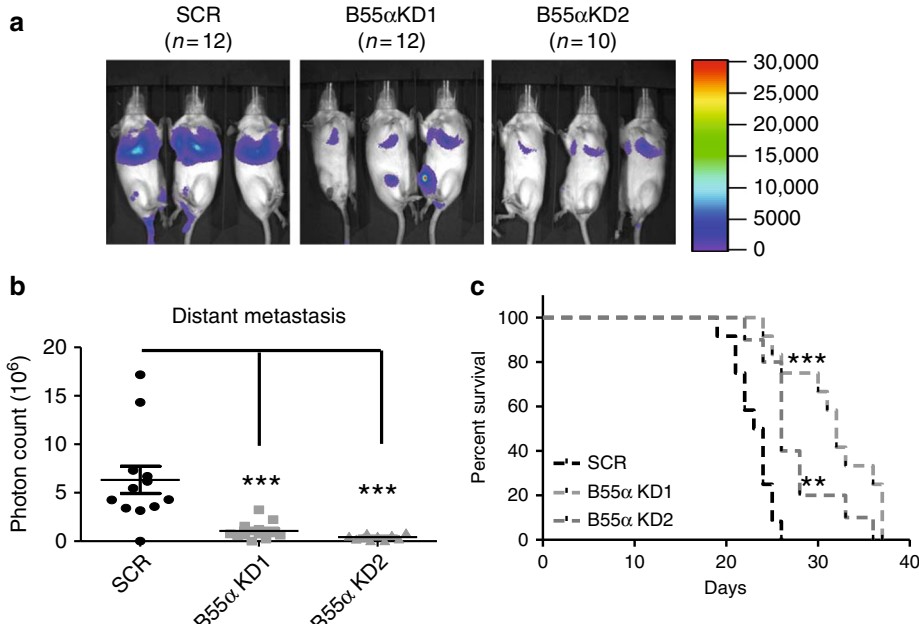

**Fig. 7** Knockdown of B55α decreases breast cancer metastasis. **a** Representative bioluminescent images of metastatic burden in mice at 3 weeks after injection is shown. **b** Quantification of photon flux in **a**, *p*-value is calculated using one-way ANOVA followed by Tukey test, ***$p < 0.001$. **c** Kaplan–Meier curves demonstrate the percentage of mice that survived, *p*-value is calculated using a log-rank test followed by Bonferroni correction, **$p < 0.01$, ***$p < 0.001$

tumor promotional role through its ability to regulate transformation, migration, invasion, DNA damage response[14,48], ERβ activity[17], and metastasis[13]. In this paper, we demonstrate that the Eya3–PP2A interaction plays a critical role in mediating c-Myc stabilization and late-stage metastasis in a breast cancer model. Given the widespread transcriptional activity of c-Myc and its multiple oncogenic roles[49], Eya proteins may contribute to tumor cell proliferation, transformation, migration, invasion, and metastasis at least in part through controlling c-Myc levels via Eya-associated Thr phosphatase activity. Moreover, as the Eya Thr phosphatase activity has been reported to mediate innate immunity[18,20], the Eya–PP2A interaction may impact tumor progression not only through regulating c-Myc levels, but also through immune modulation, potentially facilitating the tumor-promotional actions of Eya's transactivation and Tyr phosphatase activities. Finally, because the interaction with PP2A occurs in the NTD of Eya, which also contains the transactivation ability of Eya[50], it is further possible that this association affects Eya co-activator functions. Although the C-terminal Tyr phosphatase and Six-interacting domain were previously thought to be the key regions that promote metastasis[5,6], our data suggest that the N-terminal associated phosphatase activity is also important for late-stage metastasis. It is also worth noting that a previous report suggests that the C-terminal domain of Eya1 is important for Eya's Thr phosphatase activity[21]. However, under our experimental conditions, the NTD, but not the CTD, of mEya3 has significant interaction with PP2A and contains substantial Thr phosphatase activity. Because this previous report focused on human Eya1, there remains the possibility that this discrepancy reflects the different properties of different Eya family members and/or Eya from different species.

Our results also reveal Eya as a previously unrecognized regulator of PP2A, a major Thr phosphatase in cells. PP2A is typically regulated through its B subunits, which determine the substrate specificity, subcellular localization, and enzymatic activity of the holoenzyme[24]. Thus, the binding of PP2A to Eya proteins via a particular B subunit may result in altered PP2A

behavior, or may even switch PP2A from tumor suppressive to tumor promotional. Indeed, we demonstrate that Eya binds to PP2A specifically through its B55α subunit and that suppression of PP2A-B55α decreased metastatic burden in our experimental metastasis model, uncovering a tumor promotional role for PP2A-B55α in breast cancer. This observation is consistent with two recent studies, which demonstrated that the PP2A–B55α complex contributes to tumor progression through enhancing target gene occupancy of c-Jun and stimulating oncogenic signaling (ERK, AKT, and Wnt) in colorectal and pancreatic cancer models, respectively[30,31]. Thus, although PP2A is considered to be tumor suppressive, it may stimulate tumor progression when certain regulatory subunits bind to a protein, such as Eya3.

Finally, our results reveal a mechanism of regulating the stability of a major oncogene, c-Myc (Fig. 8). Previous studies have shown that PP2A dephosphorylates S62 and reduces c-Myc stability via physically interacting with c-Myc through its B56α subunit[26]. However, we demonstrate that Eya3, by binding to B55α of PP2A, leads to dephosphorylation of T58 and increased c-Myc stability, whereas Eya3 does not interact with B56γ or B56α (Fig. 4b and Supplementary Fig. 5b). Our data showing that Eya3 H79A interacts with c-Myc (Fig. 5a) indicate that the interaction between Eya3 and c-Myc does not depend on PP2A, even though these data do not prove that Eya3 directly interacts with c-Myc. Given that a previous paper suggests that Eya1 physically interacts with c-Myc in a direct manner[21], an attractive possibility is that Eya3 directly interacts with B55α, bringing PP2A to c-Myc to enable dephosphorylation of pT58 (Fig. 8). Further, our in vivo experiments demonstrate that c-Myc is stabilized by Eya3 in long-term experiments performed in animals, as the percentage of c-Myc positive regions is higher in SCR + EV and KD + Eya3[WT] lung metastases when compared to those with KD and KD + Eya3 [H79A] (Fig. 6c, d). Taken together, these data suggest that PP2A is involved in both the stabilization and destabilization of c-Myc, depending on the participation of different regulatory B subunits and the availability of Eya3. Whether B55α (in the presence of Eya3) and B56α subunits compete with

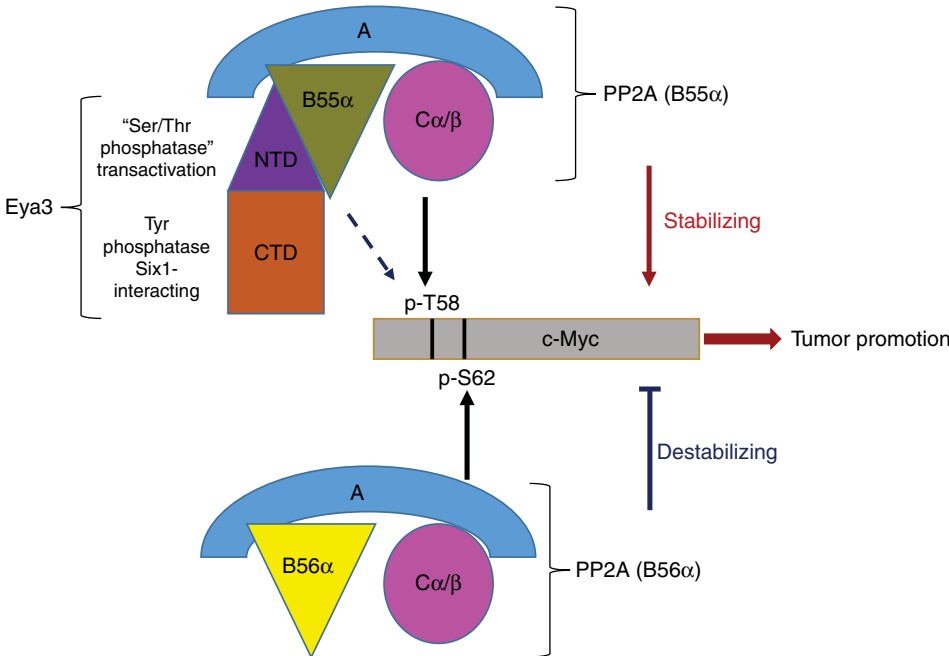

**Fig. 8** Schematic representation of the multiple functions of Eya. The N-terminal domain (NTD) of Eya3 possesses transactivating potential and "Thr phosphatase activity", while the C-terminal domain is essential for interacting with Six family members and contains Tyr phosphatase activity. In this paper, we reveal that the apparent Eya3 Thr phosphatase activity arises from an association between the Eya3 NTD and the B55α subunit of PP2A. Previous studies demonstrate that PP2A–B56α dephosphorylates pS62 and causes c-Myc degradation, resulting in tumor suppression. In contrast, we demonstrate that when Eya3 is present, the Eya3–PP2A–B55α complex dephosphorylates pT58 and stabilizes c-Myc (it is possible that Eya3 directly interacts with c-Myc and brings PP2A-B55α to c-Myc), leading to tumor promotion. Taken together, Eya3 serves as a key regulator of PP2A to modulate tumor development, likely in part via controlling c-Myc stability

each other for PP2A holoenzyme formation, or whether they behave sequentially and independently when altering the phosphorylation status of c-Myc, remains to be elucidated. Interestingly, Virshup and colleagues observed a similar B subunit-dependent PP2A substrate specificity switch, where PP2A containing a 72 kD B subunit dephosphorylates Ser120 and Ser123 in simian virus Large T antigen, leading to the stimulation of viral DNA replication, while PP2A containing a 55 kD B subunit dephosphorylates Thr124, leading to the inhibition of T antigen[51].

It is well established that c-Myc, a transcription factor that acts as a "super controller" of the genome[52,53], controls diverse biologic processes, such as cell cycle progression, apoptosis, and transformation. c-Myc overexpression or hyperactivation is among the most common drivers for human cancer and its amplification is associated with poor prognosis in multiple tumor types[53]. Although c-Myc has been viewed as a promising anticancer target, it encodes a helix–loop–helix type of transcription factor with no obvious druggable domains, making it difficult for therapeutic intervention. Understanding the molecular mechanism of how c-Myc is regulated (such as by Eya3 and PP2A) provides an alternative approach through which c-Myc may be targeted in the many tumor types in which it is implicated.

## Methods
**Cell culture**. Mammalian cells were cultured in an incubator at 37 °C with 5% CO$_2$. HEK293FT (this line has been in Ford lab since its inception, and can be obtained from ATCC PTA-5077), MCF7 (this line has been in Ford lab since its inception, and can be obtained from ATCC HTB-22), and Met1 (a kind gift from Alexander D. Borowsky) cells were cultured using Dulbecco's high glucose modified eagle medium (DMEM, Hyclone SH30022.01) supplemented with 10% fetal bovine serum (FBS, corning 35-010-CV), 2 mM L-glutamine (Hyclone SH30034.02), and 1% penicillin streptomycin (pen-strep, Hyclone SV30010), while 66cl4 (a kind gift from Fred R. Miller) cells were cultured using DMEM supplemented with 10% calf serum (Sigma C8056-500ML), 2 mM L-glutamine, 1 mM minimum essential

medium (MEM) nonessential amino acids (NEAA, Gibco 11140-50) and 1% pen-strep. MB-MDA-231 (this line has been in Ford lab since its inception, and can be obtained from ATCC HTB-26) cells were cultured in MEM/EBSS (Hyclone SH30024.01) supplemented with 5% FBS, 2 mM L-glutamine, 1 mM NEAA, 10 mM HEPES (Hyclone, SH30237.01), and 0.8 µg/ml bovine pancreatic insulin (Sigma I0516-5ML). All human cell lines (HEK293FT, MCF7, and MDA-MB-231) were authenticated by the University of Colorado Cancer Center Tissue Culture Shared Resource using STR fingerprinting in March or August of 2017.

**Protein expression and purification**. mEya3 (residues 1-526), mEya3 NTD (residues 1-240), mEya3 CTD (residues 241-526), hEya1, hEya2, hEya3, and hEya4 were subcloned into the pEF-BOS-EX vector[18] (gift of Dr. Shigekazu Nagata) with a Flag tag at the N-terminus. The mEya3$^{H79A}$, mEya3$^{Y77A}$, mEya3$^{Y90A}$, and mEya3$^{Y77AH79A}$ mutants were generated via site-directed mutagenesis[54] using PCR with Q5 polymerase from New England BioLabs Inc.

To express various Eyas (different EDs and mutations) in mammalian cells, the pEF-BOS-EX vector carrying different Eya constructs was transfected into HEK293FT cells using the CaCl$_2$ method. Cells were harvested after 48 h and incubated in lysis buffer (TBS: 50 mM Tris–HCl, pH 7.4 and 150 mM NaCl) including 1 mM EDTA, 1 mM DTT, and 0.5% TRITON X-100 for 30 min on a shaker at 4 °C. The cells were sonicated on ice 3–5 times, 20 s each time using a micro-tip, and cooled down for 1 min between sonication. The cell lysates were then centrifuged for 30 min at 18,000 × g at 4 °C, after which the supernatant was incubated with anti-Flag M2 affinity Gel (Sigma) at 4 °C overnight. The resin was washed in 1 ml TBS three times, 1 ml TBS including 250 mM NaCl another three times, and then in 1 ml TBS, including 500 mM NaCl, two more times. Proteins were then eluted using 142 ng/µl 3 × Flag peptide (Sigma) in TBS buffer (for the c-Myc and Eya interaction, the resin was washed in TBS six times.) Purified proteins were used for western blot analysis and Thr and Tyr phosphatase activity assays.

To express and purify Eya proteins from E. coli, N-terminal Flag tagged-mEya3 was subcloned into the pGEX-6P1 vector (GE Healthcare). The GST-PP2A-Aα and GST-PP2A-B56γ1 constructs were gifts from Dr. Wenqing Xu at University of Washington Seattle. The plasmids were transformed into E.coli XA-90 cells. Overnight cultures of XA90 cells containing mEya3, PP2A-Aα, and PP2A-B56γ1 expression constructs were used to inoculate 2XYT medium containing ampicillin (100 µg/ml), grown until the OD600 reached 0.6–0.8, and then induced with 0.5 mM IPTG overnight at room temperature. Cells were harvested, lysed by sonication in 50 mM Tris (pH 8.0), 250 mM NaCl, 5% glycerol, and 1 mM DTT, and purified on glutathione resin. Precession protease was used to release mEya3 proteins from the resin, followed by further purification on a Superose$^{TM}$ 6 (GE

Healthcare) column. GST-Aα and GST-B56γ1 were expressed at 18 °C with the addition of 0.5 mM IPTG and eluted from the glutathione resin using 500 mM reduced glutathione (pH 8.0) and dialyzed into 50 mM Tris (pH 8.0), 5% glycerol, and 1 mM DTT with 100 mM NaCl or 250 mM NaCl.

To express and purify the PP2A-B55α subunit in insect cells, the human PP2A-B55α subunit was subcloned from the pMIG-Flag-B55α plasmid purchased from Addgene into the pFastBac-HTB vector (Invitrogen), and transformed into DH10Bac™ cells to generate Bacmid DNA. Recombinant baculovirus was generated in insect SF9 cells using the Cellfectin® II Reagent (Thermo Fisher Scientific). His$_8$-tagged PP2A-Cα was cloned into an engineered pFastBack vector (Invitrogen). Production of baculoviruses followed a similar procedure as previously described[55]. PP2A-Bα and Cα were overexpressed in baculovirus-infected Hi-5 suspension culture. Cells were resuspended in buffer (50 mM Tris (pH 8.0), 100 mM NaCl, 1 mM DTT and 10 mM imidazole) and lysed by sonication. The protein was then purified on Ni-NTA resin (QIAGEN). After washing with 10, 20, and 40 mM imidazole, the protein was eluted using 500 mM imidazole and dialyzed into 50 mM Tris (pH 8.0), 100 mM NaCl, and 1 mM DTT.

**GST pull-down experiments**. For GST pull-down experiments, GST-PP2A-Aα subunit was expressed and purified in *E.coli*, His-B55α and His-Cα subunits were expressed and purified from insect cells. Equal amounts of different subunits were mixed to form the PP2A subcomplex AC and holoenzyme ABC. GST-PP2A-Aα, Aα + Cα, or Aα + B55α + Cα (2 μg each subunit) was mixed with 3 μg of Flag-mEya3 protein purified from *E. coli*, and incubated with the glutathione resin in 50 mM HEPES (pH 6.9), 50 mM NaCl, 4 mM cysteine, and 0.2% DMSO. After washing five times using 50 mM HEPES (pH 6.9), 50 mM NaCl, and 1 mM DTT, 1X SDS-PAGE loading dye was added to the glutathione resin and the pull down was analyzed by SDS PAGE. Ni$^{2+}$ pull down was carried out similarly, although His-B protein purified from insect cells was incubated with the Ni-NTA resin, and proteins were eluted from the resin using 50 mM Tris (pH 8.0), 100 mM NaCl, 500 mM imidazole, and 1 mM DTT before SDS-PAGE analyses.

**Thr and Tyr phosphatase activity assays**. Three hundred micrometer of pThr peptide (KRpTIRR, Millipore), pSer peptide (RRApSVA, Millipore), or pH2AX peptide (KATQASQEpY, Abgent) was incubated with 200–500 ng of Eya protein in a 25 μl phosphatase reaction. An equal amount of purified protein was used in each set of phosphatase reactions, except for hEya2 whose yield is much lower than the other Eya proteins and thus a lower quantity of protein was used in the assay (concentrations specified in figure legend). The reaction was incubated at 37 °C for 1 h after which 100 μl Malachite Green mixture was added and allowed to incubate for 15 min. Absorbance was measured at 620 nm on a Spectra-max-plus384 plate reader. All phosphatase activity assays were normalized to the amount of protein used.

**Luciferase assays**. A total of 2000–3000 MCF7 cells were plated per well (three replicate wells per condition) in 100 μl DMEM in clear 96-well plates, and the cells were allowed to attach overnight. The next morning, media was changed to 100 μl complete medium, 30 min prior to transfection. 40 ng pcDNA3.1-Six1, 30 ng pEF-BOS-EX vector (or pEF-BOS-EX-Eya3$^{WT}$ or pEF-BOS-EX-Eya3$^{H79A}$), 60 ng MEF3 luc reporter in pGL3 vector, and 14 ng Renilla vector were transfected into cells in each well following manufacturer's instructions for Fugene (Promega). Each well was replaced with fresh complete medium 16 h after transfection. Cells were lysed 48 h after transfection, and luciferase activity was measured using the Promega Dual Luciferase kit and the Turner Biosystems Modulus microplate reader.

**Mass spectrometric analyses**. Eya3 protein purified from HEK293FT cells was digested according to the FASP protocol[56] using a 10 kDa molecular weight cutoff filter. Samples were analyzed on a Q Exactive HF quadrupole orbitrap mass spectrometer (Thermo Fisher Scientific, Waltham, MA, USA) coupled to an Easy nLC 1000 UHPLC (Thermo Fisher Scientific) through a nanoelectrospray ion source. Data acquisition was performed using the instrument supplied Xcalibur™ (version 4.0) software. MS/MS spectra were extracted from raw data files and converted into mgf files using a PAVA script (UCSF, MSF, San Francisco, CA). These mgf files were then independently searched against human SwissProt database using an in-house Mascot™ server (Version 2.5, Matrix Science). Mass tolerances were ± 10 ppm for MS peaks, and ± 0.05 Da for MS/MS fragment ions.

**Antibody and western blot analysis**. Cells were harvested and lysed in RIPA buffer (150 mM NaCl, 1.0% IGEPAL® CA-630, 0.5% sodium deoxycholate, 0.1% SDS, and 50 mM Tris, pH 8.0) containing protease inhibitors (Thermo Scientific, A32965). Polyacrylamide gel electrophoresis of cell lysates (20 or 50 μg) was performed, after which proteins were transferred to PVDF or nitrocellulose membranes, and 5% nonfat milk was used to block non-specific antibody binding. Membranes were incubated with the following primary antibodies: anti-Flag IgG (Sigma, F1804, 1:3000 dilution), anti-Six1 IgG (ATLAS, HPA001893,1:1000), anti-Eya3 IgG (Bethyl Laboratories, A302-689A, 1:1000 dilution), anti-PP2A-C IgG (Cell signaling, 2038, 1:3000 dilution), anti-PP2A-B55α IgG (Santa Cruz, sc-81606, 1:500 dilution), anti-B56α IgG (Santa Cruz, sc-6116, 1:500 dilution), anti-PP2A-Aα IgG (Santa Cruz, sc-6112, 1:500 dilution), anti-c-Myc IgG (Abcam, ab32, 1:500 or

1:1000 dilution), anti-pT58 c-Myc IgG (Abm, Y011034, 1:1000 dilution), anti-pS62 c-Myc IgG (Abcam, ab78318, 1:1000 dilution), anti-β-Tubulin IgG (Sigma, T4026, 1:3000 dilution), and anti-GAPDH IgG (Cell signaling, D16H11, 1:3000 dilution). Following primary antibody incubation and washes, secondary antibodies used were as follows: anti-mouse IgG-HRP (Sigma, A9044, 1:3000–10,000 dilution) and anti-rabbit IgG-HRP (Sigma, A9169, 1:1000–3000 dilution). Uncropped western blots are shown in Supplementary Fig. 8.

**PLA**. HEK293FT, mouse or human breast cancer cells were fixed in 4% paraformaldehyde and permeabilized using 2% Triton in PBS. The PLA assay was conducted using Duolink In Situ Red Starter Kit Mouse/Rabbit (Sigma-Aldrich, DUO92101) following the manufacturer's protocol. Polyclonal anti-Eya3 IgG (Bethyl Laboratories, A302-689A), monoclonal anti-B55α IgG (Santa Cruz, sc-81606), monoclonal anti-B56α IgG (Santa Cruz, sc-136045), and monoclonal anti-PP2A-C subunit IgG (Abcam, ab33537) were used to detect the Eya3–PP2A interaction. All PLA experiments have been conducted more than three times, with at least 200 cells examined per experiment, and the graphs show quantification of representative experiments. Error bars are standard deviations. The PLA signal was calculated by determining the ratio of red fluorescence over DAPI (as a measure of cell number and size based on nuclear staining), and one-way ANOVA was used for statistical analysis.

**shRNA KD experiments**. HEK293FT cells were lentivirally transduced with pLKO.1 vectors expressing shRNAs (TRCN0000002493 and TRCN0000002483) to KD the PP2A-B55α subunit and the Cα subunit, respectively, while PP2A-Cβ subunit was knocked down via transduction of a pLKO.5 vector expressing shRNA (TRCN0000349621). In 66cl4 cells, Eya3 and the PP2A-B55α subunit were knocked down via lentivirally introducing pLKO.1 vectors expressing shRNA (TRCN0000029858) and shRNA (TRCN0000241290, TRCN0000241288), respectively.

**Generation of Eya3 addback cell lines**. The Flag-mEya3 cassette from pEF-BOS-EX was subcloned into the pMSCV vector to generate the mEya3 H79A mutant used for PLA and in vivo experiments. The wobble mutations in mEya3 are as follows: 180–200 nt in 526AA mEya3 (Accession: NP_997592.1), CCCTCGCTCATCCAATGATTA was mutated to CCCCCGATCGTCTAAC-GACTA) and H79A was mutated by changing the CAC at position 79 to GCC. All mutations were confirmed via sequencing. Mutations were generated on mEya3 in the pMSCV vector using QuickChange site-directed mutagenesis (Agilent, 200518). pMSCV vectors carrying either WT or mutated (H79A) mEya3 (all containing wobble mutations) were virally introduced into 66cl4 cells.

**c-Myc stability assay**. Fifty micrograms per milliliter of cycloheximide (Santa Cruz, sc-3508B) was added to HEK293FT or 66cl4 cells, after which cells were lysed at indicated time points up to 2 h. Western blot analysis was performed to analyze c-Myc levels, and signal was quantified using densitometry and ImageJ software. Half-lives were calculated using exponential nonlinear regression model based on the average of c-Myc levels from multiple assays ($n > 4$).

**Experimental metastasis assay**. $2.5 \times 10^6$ firefly luciferase-tagged 66cl4 control and addback cells in HBSS buffer were injected into the tail vein of Balb/c mice, and mice were euthanized once the mice were moribund. Intraperitoneal injection of 100 μl of 15 μg/ml D-luciferin (Gold Biotechnology, LUCK-1G) followed by bioluminescence imaging using Xenogen IVIS200 was conducted weekly until the first animal was euthanized. Bioluminescence signal was quantified using the Igor pro software (WaveMetrics Inc.) and statistical analysis was performed using one-way ANOVA. Kaplan–Meier survival analysis was plotted based on the dates of euthanasia and statistical analysis performed using the log-rank test followed by Bonferroni adjustment. Animal experiments were approved by the Institutional Animal Care and Use Committee of the University of Colorado Denver and Anschutz Medical Campus.

**Immunohistochemistry**. Immunohistochemistry of c-Myc was conducted by the University of Colorado Cancer Center Histology Shared Resource. Five-micron-thick paraffin sections were deparaffinized, antigen unmasked, and immunohistochemically stained for c-Myc (Abcam, Cambridge, MA; rabbit monoclonal Y69; Cat# ab32072; dilution 1:50 in TBST + 1% BSA w/v). Antigen was revealed in pH 9.5 BORG solution (Biocare Medical, Concord, CA) for 10 min at 110 °C (NxGen Decloaking chamber, Biocare) with a 10-min ambient cool down. Immunodetection was performed on the Benchmark XT autostainer (Ventana Medical Systems, Tucson, AZ) at an operating temperature of 37 °C. Primary antibody was incubated for 32 min and detected with a modified I-VIEW DAB (Ventana) detection kit. I-VIEW secondary antibody and enzyme dispensers were replaced with full-strength and half-strength (diluted in PBS, pH 7.6) polymers, respectively (Rabbit Imm-Press, Vector Laboratories, Burlingame, CA: cat# MP-7401). All sections were counterstained in Harris hematoxylin for 2 min, blued in 1% ammonium hydroxide (v/v), dehydrated in graded alcohols, cleared in xylene, and coverglass

mounted using synthetic resin. Sections were examined through bright-field microscopy and c-Myc positive regions were quantified using ImageJ software.

**Data availability**. All other relevant data are available within the Article and Supplementary Files, or available from the authors upon request.

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

## Acknowledgements

We would like to thank Drs. Aaron Patrick and Mark DellAcqua for contributions to the early development of this project. We would also like to thank Dr. Wenqing Xu at the University of Washington Seattle for providing the PP2A-Aα and PP2A-B56γ1 plasmids, the Protein Production/MoAB/Tissue Culture Shared Resource for assistance in recombinant protein production in insect cells, and the Animal Imaging Shared Resource of the University of Colorado Cancer Center (P30CA046934). We thank Dr. Monika Dzieciatkowska and the Proteomics Core Facility at the University of Colorado Denver Anschutz Medical Campus for mass spectrometry analyses, Wenxin Hu for advice on the insect cell expression system, Daniel Zheng for assistance in B56γ1 expression and purification, Deguang Kong, Joaquin Espinosa, and Ahwan Pandey for help with data analysis. This work was supported by NIH R21CA185752 (H.L.F. and R.Z.), R01CA095277 (H.L.F.), R01GM096060-01 (Y.X.), R01CA221282 (H.L.F. and R.Z.), The Front Range Cancer Challenge Grant (H.Z.), and NCI F31CA189736 (R.V.).

## Author contributions

L.Z., H.Z., R.Z., and H.L.F. conceptualized the study and designed and interpreted the experiments. L.Z., H.Z., X.L., and R.V. performed the experiments and carried out data analyses. L.Z. and H.Z. contributed equally and are listed alphabetically. M.R. and Y.X. constructed and produced the PP2A-Cα virus. P.R. and D.G. performed the statistical analysis for the mouse studies . L.Z., H.Z., R.Z., and H.L.F. wrote and edited the manuscript and all authors provided comments on this manuscript.

## Additional information

**Competing interests:** The authors declare no competing interests.

