## [Peer Review File · Nature Communications]

Reviewers' comments:

Reviewer #1 (Remarks to the Author):

This paper describes the novel finding that the previous data on Eya proteins having both Thr and Tyr phosphatase activity and capacity to dephosphorylate Myc at T58 occur through Eya protein interaction with a specific PP2A holoenzyme. The authors show that the N-terminal transactivation domain (NTD) of Eya does not actually have phosphatase function, but rather couples Eya to PP2A-B55a, which then, at least in the case of Eya3, functions on Myc to dephosphorylate T58 and increase Myc stability. They further show that mutation of Eya3 to prevent interaction with PP2A-B55a suppresses *in vivo* metastasis in a tail vein injection assay and extends survival. The data are clear, although in some places increasing replicates and cell lines used to study this mechanism would increase the robustness of the data. Further, the tumor and invasive/metastatic activity of Eya and PP2A-B55a have been previously reported, as well as the effects of Eya on stabilizing Myc, thus detracting from the novelty, except for the connection with PP2A being an intermediary, but this is not really tested in the final figure. Questions and concerns are detailed below:

1. Is the interaction between Eya3 and PP2A observed in normal cells as well as cancer cells? Does the interaction increase with metastatic potential in breast cancer cells? It looks like the antibody used in the PLA is a polyclonal anti-Eya3 IgG so I assume it would also pick up endogenous protein, however the EV condition shows almost no signal in HEK293 cells, why?
2. To show specificity for PP2A-B55a in the PLA interaction assay, does it work with Eya3 and B55a antibodies, does knockdown of B55a decrease signal, and does the B56a antibody not give a PLA signal with Eya3?
3. In figure 3b, why does the thr p'tase activity not reflect the amount of PP2A-c pulled down by the different Eya family members? It should if this is the only source of thr P'tase activity.
4. I have not seen data that PP2A-B55a only has thr and not ser P'tase activity. Please cite or test. If it has both, does this change when bound to Eya3? If not this could have implications for Ser62 phosphorylation.
5. How many times were representative PLA experiments run and are the stats shown comparing independent replicates or diversity in a single experiment? This is not clear from figure legends or methods and needs to be clearly stated what comparisons are going into the stats.
6. Figure 5a, what are Myc levels in the clonal lines expressing EV, WT or H79A Eya3? Do Myc levels go down with Eya3 KD + EV, up with +WT, down with +H79A relative to Scr EV? Data in Fig 5 is showing the same thing that was already shown in previous figures, unless Myc levels are added, it likely doesn't warrant a full figure.
7. Does the Eya3H79A mutation disrupt the interaction with B55a in addition to with PP2Ac?
8. Figure 6a, why is there not more Myc in the Eya3 WT cells (input) relative to EV or Eya3H79a as would be expected if Eya3 WT is stimulating Myc Thr58 dephosphorylation (as seen in Fig 6b)? perhaps quantification relative to GAPDH would show this?
9. Does Eya3 directly interact with Myc as suggested by the Eya3H79A IP in Fig 6a since this mutant does not interact with PP2A, PP2A would not be bridging the interaction. In this case, how does the binding of Eya3H79A directly to Myc affect the interpretation of the subsequent data? Would this affect Myc transactivation function due to interaction with MB1? What does it do to Ser62 phosphorylation levels?
10. Figure 6b needs western blot of total Myc.
11. If PP2A-B55a is helping Eya3 stabilize Myc, why does its KD not destabilize Myc relative to SCR+EV in Fig 6d? Analysis of Ser62 phosphorylation could shed light on this.
12. How does KD of Eya3, expression of Eya3H79A, and KD of B55a affect other PP2A holoenzymes, in particular PP2A-B56a? P'tase activity assay for PP2A-B56a in these conditions is important to evaluate shown effects on Myc stability.
13. Plot half-life measurements on semi-log to calculate half-life +/- SD from independent replicates.
14. Other work has suggested that the c-terminus of Eya3 has metastatic promoting function and

in the discussion the authors comment that their data argues that both the n and c terminus have important metastatic promoting activity, however, with the Eya3H79A mutant expression you see low metastatic function despite maintaining the c-terminus, could this have to do with sequestering Myc per comment 9, and that the c-terminus metastasis promoting activity requires stabilization of Myc as a downstream effector of its function? Also, what does this mutation do to the transactivation activity of Eya3? This should be examined if it is not known. Also, to examine Myc specificity in this in vivo assay rescue with the T58A Myc mutant would be important to show.

15. H&E of mets to look at number versus size of mets would be helpful. Also, to connect the mechanism to Myc in the in vivo experiments, PLA of Myc and B55a in the mets (you can do PLA on FFPE) and showing loss of Myc-B55a interaction upon Eya3 KD and with expression of the H79A mutant would add strength as well as the consequent decrease in Myc levels in mets with Eya3 KD and expression of H79A.

16. In general Figure 7 lacks a connection with the Eya3-PP2A-B55a-Myc mechanism under study, which is the novel finding in this study, as both Eya and B55a have been already linked to increased metastasis/invasion. If the H79A mutant affects the trans-activation activity of Eya3, this could also confound connecting the in vivo results to the mechanism under study. Connecting the data in this figure to the novel mechanism revealed by this work is important.

17. Better labeling on the figures themselves would enhance ease of reading.

Reviewer #2 (Remarks to the Author):

Reviewer Comments for Author

The manuscript by Zhang et al. investigated the Thr phosphatase activity of the N terminal domain of mouse and human Eya3. They found Eya3 interacts with the PP2A complex in vitro and in vivo. The authors further show that the N-terminal domain of Eya3 physically interacts with B55a subunit of PP2A and the Thr phosphatase activity of Eya3 requires PP2A. They also showed Eya3 physically interacts with c-Myc and that Eya3 and PP2A subunit B55a are required for c-Myc stability via the dephosphorylation of residue T58 of c-Myc. Finally the authors show that knockdown of Eya3 or B55a can reduce metastasis of injected 66cl4 cells, showing a biological significance of the Eya3 and PP2a phosphatase activity. The strengths of the paper are a highly novel, mechanistic finding concerning an important protein that is mostly well supported by the data presented. A major weakness of the paper, however, is conflicting data/interpretation regarding the interaction of Eya3 with PP2A and c-Myc, the role of the H79A mutation in Eya3 in this process, and the model for how Eya3 regulates c-Myc stability via PP2A.

The paper is well written and the findings would benefit the field and a wider audience interested in cancer research. The work is generally convincing and supported by controls. However, there are some concerns:

Major concerns:

1. In Figure 1G, the authors showed that a knock down of the PP2A C subunits led to decreased Eya3 Thr phosphatase activity. However, there were no controls to show that the isolated Eya3 levels from knocked down cells are the same as that of the scrambled siRNA control. The authors need to include this control.
2. Statistical analyses are missing from all of the phosphatase activity assays.
3. Where is hEya2 in Figure 3A? The "Input" is virtually non-existent. How can that be IPed and pull down more PP2A than any of the others. This needs a much better explanation.
4. Does Eya3 directly interact with/control the levels of c-Myc or is this done via PP2A? If the

latter, then we would expect H79A to not interact with c-Myc since the H79A mutation abolished Eya3 association with PP2A, but H79A does interact with c-Myc nearly as well as WT Eya3 (Fig. 6A). This would suggest that Eya3 directly interacts with c-Myc, independently of PP2A/B55. However, the authors state that "Eya3 could not stabilize c-Myc in the absence of B55a" and that their "data suggest that Eya3 controls c-Myc levels through dephosphorylating T58 via a mechanism that requires PP2A-B55a." Are the authors suggesting that although Eya3 directly interacts with both B55 and c-Myc, these interactions are independent of each other and that Eya3 acts to bring c-Myc and PP2A together to dephosphorylate c-Myc? It seems not to be the case as there is no such model presented and no direct interaction of Eya3 and c-Myc is depicted in the Figure 8 model. These results therefore appear to be at odds with each other and their presented model and this needs to be addressed.

Minor concerns:

1. Figure 1H graph title has a typo.
2. In Figure 2B, Eya3H79A appears to have significantly more PLA signal than the empty vector control while the 2C shows they have similar PLA / DAPI levels. The authors should account for this discrepancy in their interpretation of their data.
3. In Figure 4A, the blot and the gel do not match. A MW marker (3rd lane from the right) is unlabeled in the gel and should be labeled. while it is missing in the blot.
4. In Figure 4G, there is a formatting error in the Eya3 KD label, the KD part is embedded within the fluorescence image.
5. Figure 5 shows the knockdown and transfection is working. This figure is more suited to be a supplemental figure instead of a main figure.
6. In the main text, line 226, the Figure 6 reference is mislabeled. It should refer to Figure 6b, left panel instead of right panel.
7. The first sentence of the Introduction is incorrect; the *eya* gene was first discovered as being required for normal eye development in *Drosophila* (Bonini et al, Cell 72, 379-395, 1993).
8. Figure 6 Legend: for (b) it refers to 'upper' and 'lower' panels but it should say 'left' and 'right'
9. Why was His-B55a and His-Ca expressed and purified from insect cells and not from *E. coli* as with GST-Aa?
10. Although the authors acknowledged the facility for performing the mass spectrometry, the mass spectrometry protocol and reagents used are missing from the methods section. The rest of the mass spectrometry data is also missing from the manuscript.
11. In Figure 6B – C, the blots showing c-Myc levels are faint are difficult to see. Better images are needed.

We would like to thank the reviewers for their constructive comments and helpful suggestions. We have incorporated these comments into our revised manuscript, and outline below our point-by-point responses (in blue) to reviewers' comments and questions (in black). Major changes in the manuscript in response to reviewers' comments are highlighted in yellow within the text. We believe that these changes have significantly strengthened our manuscript, and look forward to hearing back from the reviewers and senior editor regarding our revised manuscript.

Reviewer #1:

The data are clear, although in some places increasing replicates and cell lines used to study this mechanism would increase the robustness of the data.

We apologize to the reviewer if the number of replicates was not made clear in our manuscript. We have carried out all experiments in the manuscript at least three times with the exception of animal experiments which were carried out twice, and this is now clearly stated in our methods as well as in the figure legends. In addition, we have utilized both breast cancer and human embryonic kidney cells to demonstrate that this mechanism holds true across both cancerous and normal cells. Finally, we added a second KD of B55 α to all data in which we performed B55 α KD (see Fig. 6, supplementary Fig. 3b-f and supplementary Fig. 6a).

Further, the tumor and invasive/metastatic activity of Eya and PP2A-B55a have been previously reported, as well as the effects of Eya on stabilizing Myc, thus detracting from the novelty, except for the connection with PP2A being an intermediary, but this is not really tested in the final figure.

To our knowledge, PP2A-B55 α has never before been shown to mediate breast cancer metastasis. In addition, we are the first to demonstrate that Eya3 stabilizes Myc in the context of breast cancer. Further, significant novelty in this study lies in the demonstration that Eya proteins (Eya3 or other Eyas) actually regulate Myc stability via a Thr phosphatase activity that is NOT intrinsic to Eya, but rather through an association with PP2A, as published data suggest that Eya proteins act as intrinsic Thr phosphatases. Furthermore, it is novel to demonstrate that Eya3 is regulating PP2A function in a way that changes which residue it dephosphorylates on Myc, at least in part by recruiting a different PP2A regulatory subunit to dephosphorylate T58 compared to the one that is recruited to dephosphorylate S62. Taken together, there is significant novelty in this study, and it will change not only our understanding of how the Eya3 phosphatase works, but further identifies a completely new regulator of the most abundant phosphatase in the cell, PP2A. Finally, given that we show that the Eya3 point mutant that cannot bind PP2A, does not stabilize Myc, nor mediate metastasis in vivo, we have in fact made the connection that PP2A interaction with Eya3 is required for the phenotypes observed.

1. Is the interaction between Eya3 and PP2A observed in normal cells as well as cancer cells? Does the interaction increase with metastatic potential in breast cancer cells? It looks like the antibody used in the PLA is a polyclonal anti-Eya3 IgG so I assume it would also pick up endogenous protein, however the EV condition shows almost no signal in HEK293 cells, why?

Yes, the interaction occurs in both cancer and non-cancerous cells such as the 293FT cells (see Figs. 2b and 2c in the revised manuscript). To address the question of whether there is a correlation of metastatic potential and the Eya3-PP2A interaction, we examined the Eya3-PP2A-C interaction in two different sets of isogenic cell lines, one set derived from a spontaneous mammary tumor in mice, the 67NR (non-metastatic) and 66cl4 (metastatic) cells¹, and another set derived from an MMTV-PyMT transgenic mammary tumor². At least in these two sets of isogenic cell lines, we were unable to observe a clear increase in the Eya3-PP2A interaction when the metastatic variants (66cl4 and Met1) were compared to their non-metastatic counterparts (67NR and DB-7 respectively)^{1,3} (see Rebuttal Fig. 1).

In HEK293FT cells, endogenous Eya3-PP2A interaction could be detected in the EV group. However, the PLA signal in the WT Eya3 overexpressing group would be saturated if we used the exposure time needed to observe the endogenous interaction. Thus, exposure time, brightness and contrast settings were applied similarly to both sets, and signal can only be observed in the Eya3 overexpressing group. However, if we use a

longer exposure time (which saturates the signal in 293FT cells overexpressing Eya3), we can observe clear signal in the endogenous setting (see Rebuttal Fig. 2, below).

2. To show specificity for PP2A-B55a in the PLA interaction assay, does it work with Eya3 and B55a antibodies, does knockdown of B55a decrease signal, and does the B56a antibody not give a PLA signal with Eya3?

Yes, PLA also works with Eya3 and PP2A-B55 α antibodies, and KD of B55 α reduces signal (new data added as Supplementary Fig. 3c&d). In contrast, no Eya3-PP2A-B56 α interaction was detected in the 66cl4 lines (see data below in Rebuttal Fig. 3 and new Supplementary Fig 4b.)

3. In figure 3b, why does the thr p'tase activity not reflect the amount of PP2A-C pulled down by the different Eya family members? It should if this is the only source of thr P'tase activity.

The phosphatase activity in Fig. 3b was normalized to the amount of protein used in the reaction, so it reflects the activity from equal amounts of protein. As can be seen in Fig. 3a, even with small amounts of Eya2 pulled down in the IP, the amount of PP2A which co-precipitates is equivalent to the amount pulled down with the other Eya family members (where the IP contains far more Eya). These data suggest that Eya2 may bind PP2A more efficiently, and explain why its phosphatase activity, after being normalized to the protein level, is much higher than the other three (Fig. 3b).

4. I have not seen data that PP2A-B55a only has thr and not ser P'tase activity. Please cite or test. If it has both, does this change when bound to Eya3? If not this could have implications for Ser62 phosphorylation.

PP2A-B55 α has both Thr and Ser Phosphatase activity, although its Thr phosphatase activity is stronger (see new Supplementary Fig. 1b). Eya3 bound to PP2A has much higher Thr than Ser phosphatase activity (compare Supplementary Fig. 1a and 1b). These data suggest that binding of Eya3 to PP2A changes its preference for different substrates (at least when utilizing these specific peptides).

5. How many times were representative PLA experiments run and are the stats shown comparing independent replicates or diversity in a single experiment? This is not clear from figure legends or methods and needs to be clearly stated what comparisons are going into the stats.

All PLA experiments were conducted three or more times and the graphs shown are quantification of the experiments shown (which are representative of all replicates). Thus, error bars are standard deviations. We have now stated this more clearly in both the figure legends and in the methods.

6. Figure 5a, what are Myc levels in the clonal lines expressing EV, WT or H79A Eya3? Do Myc levels go down with Eya3 KD + EV, up with +WT, down with +H79A relative to Scr EV? Data in Fig 5 is showing the same thing that was already shown in previous figures, unless Myc levels are added, it likely doesn't warrant a full figure.

As suggested by the reviewer, the original Figure 5 was moved and is now Supplementary figure 5. In the revised Fig.5b, we now include the total Myc levels. Indeed, total Myc levels go down in Eya3 KD + EV, up with +WT, and down with +H79A relative to SCR+EV. The results thus agree with the reviewer's prediction.

7. Does the Eya3 H79A mutation disrupt the interaction with B55a in addition to with PP2A-c?

Yes, the Eya3^{H79A} mutation disrupts the interaction with B55 α (please see Fig. 1d and Supplementary Fig. 5b&c).

8. Figure 6a, why is there not more Myc in the Eya3 WT cells (input) relative to EV or Eya3H79A as would be expected if Eya3 WT is stimulating Myc Thr58 dephosphorylation (as seen in Fig 6b)? Perhaps quantification relative to GAPDH would show this?

Observing the alterations in total Myc levels at steady state is more difficult than observing half-life differences or changes in T58 phosphorylation, likely as production and turnover of Myc is very dynamic. Nonetheless, if the data from the Western blots shown in Fig. 5a are normalized to the loading control (GAPDH), the Myc level is indeed higher in the presence of WT Eya3 when compared to the EV and Eya3H79A conditions (see Rebuttal Fig. 4 below).

9. Does Eya3 directly interact with Myc as suggested by the Eya3H79A IP in Fig 6a since this mutant does not interact with PP2A, PP2A would not be bridging the interaction. In this case, how does the binding of Eya3H79A directly to Myc affect the interpretation of the subsequent data? Would this affect Myc transactivation function due to interaction with MB1? What does it do to Ser62 phosphorylation levels?

The fact that Eya3^{H79A} IP brings down Myc indicates that PP2A is not mediating the Eya3 and Myc interaction, but does not prove that Eya3 and Myc directly interact, since their interaction can be mediated by proteins other than PP2A in the cell lysate. However, given that previous work demonstrated that Eya1 interacts directly with c-Myc⁴, Eya3 likely also binds directly to Myc, potentially serving to recruit PP2A-B55 α to Myc. Although we do not know whether this affects MYC transactivation function due to influencing MB1, c-Myc stabilization will affect Myc mediated transcription regardless of whether there is an additional influence through MB1. In addition, we have found that binding of Eya3 to Myc not only affects the phosphorylation

status of T58 (dephosphorylating it), but also increases pS62 phosphorylation (we have now included this data in Fig. 5b). Thus, our data suggest that Eya3 is switching the preference of PP2A to pT58 rather than pS62.

10. Figure 6b needs western blot of total Myc.

We have added total Myc to this figure, which is now the revised Fig. 5b.

11. If PP2A-B55a is helping Eya3 stabilize Myc, why does its KD not destabilize Myc relative to SCR+EV in Fig 6d? Analysis of Ser62 phosphorylation could shed light on this.

For the sake of easy visualization, exposure times were not the same for all blots in the original manuscript. Blot intensity was adjusted to a similar level to allow easy visualization and estimation of degradation trends in different conditions. However, we have now applied the same exposure time to these figures which indeed shows that suppression of B55 α decreases c-Myc level (see revised Fig 5e).

Please see answer to comment #9 above to address the Ser62 phosphorylation changes with Eya3 KD (now included in Fig. 5b).

12. How does KD of Eya3, expression of Eya3H79A, and KD of B55a affect other PP2A holoenzymes, in particular PP2A-B56a? P'tase activity assay for PP2A-B56a in these conditions is important to evaluate shown effects on Myc stability.

We demonstrate that the level of PP2A-B56 α did not change significantly under Eya3 or B55 α KD conditions (see below, Rebuttal Fig. 5 and Supplementary Fig. 4a). Since the PP2A-B56 α level was not changed and Eya3 does not interact with B56 α (Supplementary Fig.4b), our data do not suggest that these conditions would alter the phosphatase activity of PP2A-B56 α . Furthermore, isolating enough endogenous PP2A-B56 α with sufficient purify out of mammalian cells for activity assays will be technically challenging, and we believe it is thus out of the scope of this manuscript.

13. Plot half-life measurements on semi-log to calculate half-life +/- SD from independent replicates.

We have plotted the half-lives on a semi-log scale as suggested (see Revised Figs. 5d & f).

14. Other work has suggested that the c-terminus of Eya3 has metastatic promoting function and in the discussion the authors comment that their data argues that both the n and c terminus have important metastatic promoting activity, however, with the Eya3H79A mutant expression you see low metastatic function despite maintaining the c-terminus, could this have to do with sequestering Myc per comment 9, and that the c-terminus metastasis promoting activity requires stabilization of Myc as a downstream effector of its function? Also, what does this mutation do to the transactivation activity of Eya3? This should be examined if it is not known. Also, to examine Myc specificity in this in vivo assay rescue with the T58A Myc mutant would be important to show.

It is possible that both the C-terminus and N-terminus play overlapping roles in tumor progression through influences on Myc, as suggested by the reviewer. However, it is also possible that the two ends of the molecule independently influence metastasis via affecting Myc as well as additional pathways. For example, it was previously suggested that the C-terminal tyrosine phosphatase activity of Eya3 mediates metastasis via influences on the actin cytoskeleton and activation of Rac/cdc42⁵. It may be that the two ends of the molecule are both necessary for the full effect of Eya on metastasis via influencing either overlapping or differing pathways, and that removal of either activity dramatically influences the overall function of Eya to mediate metastasis.

Importantly, the H79A mutant of Eya3 does not diminish transcription mediated by the Six1/Eya3 complex (Supplementary Fig. 2c). In fact, it may even be enhanced.

Indeed, performing a rescue experiment with T58A c-Myc mutant addback in our cell lines would underscore the importance of c-Myc in Eya3 Ser/Thr phosphatase mediated metastasis. However, we are not arguing that the sole effects of the Eya3 Thr phosphatase are through regulating Myc. While control of Myc may be one means by which Eya affects metastasis, it is highly likely that the Eya3 Thr phosphatase influences metastasis via additional means, such as via effects on the immune system. In fact, we currently have a paper in revision with *J Clin Invest* that demonstrates a role for the Eya3 Thr phosphatase in regulating the tumor immune microenvironment, and thus we feel that making a Myc point mutation is out of the scope of this article, as we likely have to examine several contributors to the metastatic phenotype downstream of the Eya3 Thr phosphatase. In addition, this experiment is technically difficult as our addback lines already contain three constructs: firefly luciferase, shEya3 and Eya3 addback (with a wobble mutation). In order to test the specificity of Myc in this model, we would need to make stable lines in which we add in two additional constructs (a Myc KD and add-back construct), which may alter cellular phenotypes.

15. H&E of mets to look at number versus size of mets would be helpful. Also, to connect the mechanism to Myc in the in vivo experiments, PLA of Myc and B55a in the mets (you can do PLA on FFPE) and showing loss of Myc-B55a interaction upon Eya3 KD and with expression of the H79A mutant would add strength as well as the consequent decrease in Myc levels in mets with Eya3 KD and expression of H79A.

We euthanized mice at different time points during the experiment (once they were morbid), in order to obtain a survival curve for the manuscript. As such, we cannot fairly compare the number or size of metastases across groups by H&E as they are all taken at the endpoint. We have performed PLA of Myc and B55 α on lung sections, but the results are inconclusive since the PLA signals are weak and almost all in the cytoplasm (they would be expected to occur in the nucleus also). We were able to detect higher levels of c-Myc in SCR+EV and KD+Eya3^{WT} lung metastases when compared to Eya3 KD or KD+Eya3^{H79A} (new Supplementary Fig. 7), indicating Eya3 Ser/Thr phosphatase stabilizes c-Myc in metastases.

16. In general Figure 7 lacks a connection with the Eya3-PP2A-B55a-Myc mechanism under study, which is the novel finding in this study, as both Eya and B55a have been already linked to increased metastasis/invasion. If the H79A mutant affects the trans-activation activity of Eya3, this could also confound connecting the in vivo results to the mechanism under study. Connecting the data in this figure to the novel mechanism revealed by this work is important.

Indeed, Eya and B55 α have both independently been reported to induce metastasis. However, to our knowledge, PP2A-B55 α has not been reported to *enhance breast cancer metastasis*. Further, the H79A condition, which represents Eya3 that cannot bind PP2A, has never before been tested for its effects on metastasis (demonstrating that the INTERACTION between Eya3 and PP2A is required for the metastatic effects). Moreover, our group and others have shown the importance of Six-interacting domain and Tyr phosphatase activity of Eyas in breast cancer metastasis, while the role of Ser/Thr phosphatase activity of Eya3 in cancer remained to be investigated. In this study, we focus on identifying the novel interaction between Eya3 and PP2A, and demonstrating the importance of this Eya3 **associated** Ser/Thr phosphatase (from two directions, both through the H79A mutant, and through knocking down B55 α) in breast cancer metastasis. Since the H79A mutant does not make Eya3 transcription worse (see answer to comment #14 and Supplementary Fig. 2c), we would argue that this interaction with PP2A is important, as shown in the manuscript. As such, this finding is indeed novel in three major ways. First, it is the first manuscript to report that Eya proteins do NOT have intrinsic Thr phosphatase activity in their N-terminus, and that it is instead mediated by PP2A. Second, it is the first demonstration that Eya3 can control PP2A specificity for different amino acids on a key target, MYC. Third, it is the first demonstration that the associated Thr phosphatase activity of Eya3 plays a role in any cancer, and specifically in metastasis of breast cancer.

17. Better labeling on the figures themselves would enhance ease of reading.

Thank you for this suggestion, we have re-labeled figures in a way that we hope makes it easier to understand the figures.

Reviewer #2:

Major concerns:

1. In Figure 1G, the authors showed that a knock down of the PP2A-C subunits led to decreased Eya3 Thr

phosphatase activity. However, there were no controls to show that the isolated Eya3 levels from knocked down cells are the same as that of the scrambled siRNA control. The authors need to include this control.

The amount of purified Eya3 proteins from the three cell lines is similar, but due to the knockdown, less PP2A-C subunits immunoprecipitate with Eya3, leading to decreased Eya3 Thr phosphatase activity (all phosphatase activities are normalized to the amount of Eya3 protein used). We have now included this data in the figure as requested (see Fig. 1g, left panel).

2. Statistical analyses are missing from all of the phosphatase activity assays.

Statistical analyses and significance level have been added to each figure as requested.

3. Where is hEya2 in Figure 3A? The "Input" is virtually non-existent. How can that be IPed and pull down more PP2A than any of the others. This needs a much better explanation.

When the Eyas are overexpressed in HEK293T cells, hEya2 did not express as well as the others, and exposure times were adjusted as to show all Eyas without saturating the signal. Nonetheless, hEya2 can be seen as a faint band in the "Input" in the current figure and is more obvious if we expose the gel longer. Unfortunately, the expression level of hEya2 is always much lower than other Eyas, even though the different Eyas are all expressed from the same promoter on the same vector. After pulling down Eya2 using an anti-Flag antibody, the hEya2 is enriched on the resin. Our experiments show that human Eya2, even though there is less of it, can pull down PP2A more efficiently, potentially because Eya2 interacts with PP2A more tightly than the other Eyas (see Fig. 3a). Although we needed to use less protein in the Eya2 phosphatase assay (as we could not pull down the same amount of protein, and thus used 50ng of Eya2 protein in the assay, and 200ng of all other Eya proteins, we normalized the phosphatase activity in Fig. 3b to the amount of Eya protein used. Thus, our data show that Eya2 phosphatase activity is much higher than that of other Eyas, likely due to increased affinity for PP2A as suggested by our pull down experiment (Fig. 3a).

4. Does Eya3 directly interact with/control the levels of c-Myc or is this done via PP2A? If the latter, then we would expect H79A to not interact with c-Myc since the H79A mutation abolished Eya3 association with PP2A, but H79A does interact with c-Myc nearly as well as WT Eya3 (Fig. 5a). This would suggest that Eya3 directly interacts with c-Myc, independently of PP2A/B55. However, the authors state that "Eya3 could not stabilize c-Myc in the absence of B55 α " and that their "data suggest that Eya3 controls c-Myc levels through dephosphorylating T58 via a mechanism that requires PP2A-B55 α ." Are the authors suggesting that although Eya3 directly interacts with both B55 and c-Myc, these interactions are independent of each other and that Eya3 acts to bring c-Myc and PP2A together to dephosphorylate c-Myc?

Our data showing that Eya3 H79A interacts with c-Myc indicate that the interaction between Eya3 and c-Myc does not depend on PP2A. However, this data does not prove that Eya3 directly interacts with PP2A, since this interaction can still be mediated by other cellular proteins present in the pull down. Given that a previous paper⁴ suggests that Eya1 physically interacts with c-Myc in a direct manner, Eya3 may also directly interact with c-Myc. An attractive possibility, as the reviewer suggested, is that Eya3 directly interacts with both B55

and c-Myc, these interactions are independent of each other and that Eya3 acts to bring c-Myc and PP2A together to dephosphorylate c-Myc. We have now depicted this possibility in the model in Figure 7 and added this possibility to the discussion.

Minor concerns:

1. Figure 1H graph title has a typo.

We thank the reviewers for pointing this out and have corrected the typo.

2. In Figure 2B, Eya3H79A appears to have significantly more PLA signal than the empty vector control while the 2C shows they have similar PLA / DAPI levels. The authors should account for this discrepancy in their interpretation of their data.

In our examination of the images, the quantitation is reflective of the representative figures. We now brought the intensity down on all figures equally as the Eya3 WT was over-saturated.

3. In Figure 4A, the blot and the gel do not match. A MW marker (3rd lane from the right) is unlabeled in the gel and should be labeled. while it is missing in the blot.

We appreciate the reviewer catching this oversight, and have appropriately labeled the figure and blot now.

4. In Figure 4G, there is a formatting error in the Eya3 KD label, the KD part is embedded within the fluorescence image.

Thank you for noticing this error. It has been corrected.

5. Figure 5 shows the knockdown and transfection is working. This figure is more suited to be a supplementary figure instead of a main figure.

As suggested, we have moved it to supplementary figure 5.

6. In the main text, line 226, the Figure 6 reference is mislabeled. It should refer to Figure 6b, left panel instead of right panel.

Thank you for noticing this issue. This is now Figure 5b in the revised manuscript and we have made sure that we referenced the correct figure.

7. The first sentence of the Introduction is incorrect; the *eya* gene was first discovered as being required for normal eye development in *Drosophila* (Bonini et al, Cell 72, 379-395, 1993).

We have corrected this and this reference now appears in the first sentence of the Introduction.

8. Figure 6 Legend: for (b) it refers to 'upper' and 'lower' panels but it should say 'left' and 'right'.

This is now Figure 5b in the revised manuscript and we have made sure that we referred the correct panel in figure legend.

9. Why was His-B55 α and His-C α expressed and purified from insect cells and not from *E. coli* as with GST-A α ?

We attempted bacterial expression for His-B55 α , but the protein was either not soluble or not functional in *E. coli*. Thus, we had to move to insect cells, which enabled successful expression and purification of His-B55 α and His-C α . This issue is discussed now in the text.

10. Although the authors acknowledged the facility for performing the mass spectrometry, the mass spectrometry protocol and reagents used are missing from the methods section. The rest of the mass spectrometry data is also missing from the manuscript.

We have added the mass spectrometry protocol in the Methods. The entire mass spectrometry data are presented in the supplementary excel file named "MS data".

11. In Figure 6B – C, the blots showing c-Myc levels are faint are difficult to see. Better images are needed.

We adjusted the brightness and contrast in the blots to provide better images (these are now shown in figure 5).

Reference

1. Aslakson, C.J. & Miller, F.R. Selective events in the metastatic process defined by analysis of the sequential dissemination of subpopulations of a mouse mammary tumor. *Cancer Res* **52**, 1399-405 (1992).
2. Hollern, D.P. & Andrechek, E.R. A genomic analysis of mouse models of breast cancer reveals molecular features of mouse models and relationships to human breast cancer. *Breast Cancer Res* **16**, R59 (2014).
3. Borowsky, A.D. et al. Syngeneic mouse mammary carcinoma cell lines: two closely related cell lines with divergent metastatic behavior. *Clin Exp Metastasis* **22**, 47-59 (2005).
4. Li, J. et al. EYA1's conformation-specificity in dephosphorylating phosphothreonine in Myc and its activity on Myc stabilization in breast cancer. *Mol Cell Biol* (2016).
5. Pandey, R.N. et al. The Eyes Absent phosphatase-transactivator proteins promote proliferation, transformation, migration, and invasion of tumor cells. *Oncogene* **29**, 3715-22 (2010).

We hope these revisions have addressed the reviewers' concerns, and look forward to hearing back from Nature Communications soon.

REVIEWERS' COMMENTS:

Reviewer #1 (Remarks to the Author):

The author's manuscript entitled "Eya3 partners with PP2A to induce c-Myc stabilization and tumor progression" brings to light an important connection between Eya3 and PP2A and highlights a potential pro-tumorigenic role for Eya3 with PP2A in cancer. I believe the authors have addressed my concerns and recommend that this manuscript be accepted for publication, with the following minor changes and suggestions:

1. Thank you for including the adjusted intensity in Rebuttal Figure 2. You may consider putting this figure in your supplemental text to demonstrate that these interactions occur in normal tissues with endogenous proteins.

2. While the statistics on the PLA experiments provide clarity for the variation between experiments, please include the number of cells that were quantified per experiment.

3. The half-life quantifications beside the images are consistent with the expression levels of Myc and the role of Eya3 and B55a in stabilizing Myc, however the line graphs (panel d and f) do not reflect all the difference in half-life shown in the quantification. Please check that these graphs are updated and correct. If so, please comment as to why there are no differences in the slopes of the lines between SCR+EV and B55a KD+EV.

4. The inclusion of Myc IHC in the lung tumors is a nice addition to the story. While Eya3 has functions independent of Myc, the majority of this publication is focused on the regulation of Myc by the Eya3-B55a interaction. You may want to consider adding the Myc expression to the main figures.

5. There are multiple places in your rebuttal where you suggest that Eya3 switches the preference of PP2A to pT58 from pS62. While Eya3 may be involved in recruiting B55a to Myc, you have not shown that the interaction between B55a or PP2A and Myc is altered with a loss of Eya3 or B55a. The complex interactions between PP2A and the B subunits suggest that these events could have reciprocity or occur simultaneously, as suggested in your discussion. The aberrant presence of Eya3 in cancer may increase B55a-Myc interactions, but I would refrain from saying that this alters the preference/affinity of PP2A towards a specific site without further data since different holoenzymes are apparently involved.

Reviewer #2 (Remarks to the Author):

The manuscript by Zhang et al. investigated the Thr phosphatase activity of the N terminal domain of mouse and human Eya3. They found the Thr phosphatase activity of Eya3 is not intrinsic but requires interactions with the PP2A complex in vitro and in vivo. The authors then show that Eya3 associates with c-Myc and Eya3 and PP2A are required for c-Myc stability via the dephosphorylation of residue T58 of c-Myc. In response to the reviews, the authors have included several requested controls and additional data to strengthen their findings and interpretations.

In our previous review, we noted a number of small mistakes in the figures, legends, and main text. These have been corrected in this revised manuscript. The missing mass spectrometry protocol and results are also now included in this manuscript as requested. We also noted that they needed additional controls and statistical analyses with their results and these have been added. To address our concern of the discrepancy between the presented results and their model, the authors have now updated their model to include the observation that hEya3 interacts with c-Myc independently of PP2A.

In addition, we had a concern with the presented hEya2 IP experiment, where the input control was nearly nonexistent. Despite the low amount of input, hEya2 can still be IP'ed. We accept their explanation that the IP hEya2 proteins are more concentrated than the input control. Although there was a small amount of pulled down hEya2, the Thr phosphatase activity of the pulled down hEya2 was very high (normalized to Eya protein levels). The authors noted that hEya2 can co-IP PP2A much more efficiently than the other hEya proteins and they attributed this to the high Thr phosphatase activity of hEya2. These are now included in the main text of the manuscript and these explanations adequately address our concerns.

In summary, we find the authors have addressed all our concerns from the initial review. The findings of this revised manuscript would benefit the field and a wider audience interested in cancer research.

Reviewer #1:

1. Thank you for including the adjusted intensity in Rebuttal Figure 2. You may consider putting this figure in your supplemental text to demonstrate that these interactions occur in normal tissues with endogenous proteins.

Thank you for this suggestion. We have added this as Supplemental Fig. 4a in our manuscript.

2. While the statistics on the PLA experiments provide clarity for the variation between experiments, please include the number of cells that were quantified per experiment.

We analyzed at least 200 cells per condition in the PLA experiments. We have added this information to both the figure legend and to the methods section of the paper.

3. The half-life quantifications beside the images are consistent with the expression levels of Myc and the role of Eya3 and B55a in stabilizing Myc, however the line graphs (panel d and f) do not reflect all the difference in half-life shown in the quantification. Please check that these graphs are updated and correct. If so, please comment as to why there are no differences in the slopes of the lines between SCR+EV and B55a KD+EV.

The observation that the lines did not appear to match the calculated half-lives is indeed correct, and we appreciate the reviewer having pointed this out. Previously, we used one phase decay equations to calculate the half-lives in this experiment. Given the disconnect between the lines on the graph and the calculated half-lives, we re-evaluated the half-lives using exponential non-linear regression in Graphpad PRISM 5.0. We feel that this type of analysis better fits the data, and have included the new analysis and data in this revised version (where the calculated half-lives appear more in line with the graphs).

4. The inclusion of Myc IHC in the lung tumors is a nice addition to the story. While Eya3 has functions independent of Myc, the majority of this publication is focused on the regulation of Myc by the Eya3-B55a interaction. You may want to consider adding the Myc expression to the main figures.

We agree with the reviewer that these data are important and highly supportive of the regulation of Myc through the Eya3-B55a interaction. Thus, we have decided to add the Myc IHC to Fig. 6 as panel (c) and (d), and as a result, have moved the B55a KD experiment into its own figure (now Fig. 7), and the original Fig. 7 is now Fig. 8.

5. There are multiple places in your rebuttal where you suggest that Eya3 switches the preference of PP2A to pT58 from pS62. While Eya3 may be involved in recruiting B55a to Myc, you have not shown that the interaction between B55a or PP2A and Myc is altered with a loss of Eya3 or B55a. The complex interactions between PP2A and the B subunits suggest that these events could have reciprocity or occur simultaneously, as suggested in your discussion. The aberrant presence of Eya3 in cancer may increase B55a-Myc interactions, but I would refrain from saying that this alters the preference/affinity of PP2A towards a specific site without further data since different holoenzymes are apparently involved.

We have checked all of the text within this manuscript, so that we are careful not to suggest that there is a "preference switch" for PP2A in the presence of Eya3 (as the reviewer correctly states that we have not formally addressed this issue).

Reviewer #2:

Reviewer #2 had no remaining concerns and stated “In summary, we find the authors have addressed all our concerns from the initial review. The findings of this revised manuscript would benefit the field and a wider audience interested in cancer research”.

We thank this reviewer also.

In closing, we hope these revisions have addressed the reviewer’s remaining concerns, and look forward to hearing back from *Nature Communications* soon.

Sincerely,

Heide L. Ford, Ph.D., Professor Department of Pharmacology
University of Colorado School of Medicine
David and Margaret Turley Grohne Chair in Basic Cancer Research
Associate Director of Basic Research, University of Colorado Cancer Center

Rui Zhao, Ph.D., Associate Professor
Department of Biochemistry and Molecular Genetics